# Towards Understanding Camera Motions in Any Video

**Zhiqiu Lin**[1*]  **Siyuan Cen**[2*]  **Daniel Jiang**[1]  **Jay Karhade**[1]  **Hewei Wang**[1]
**Chancharik Mitra**[1]  **Tiffany Ling**[1]  **Yuhan Huang**[1]  **Rushikesh Zawar**[3]
**Xue Bai**[3]  **Yilun Du**[4]  **Chuang Gan**[5]  **Deva Ramanan**[1]
[1]CMU  [2]UMass Amherst  [3]Adobe  [4]Harvard  [5]MIT-IBM

## Abstract

We introduce CameraBench, a large-scale dataset and benchmark designed to assess and improve camera motion understanding. CameraBench consists of $\sim$3,000 diverse internet videos, annotated by experts through a rigorous multi-stage quality control process. One of our core contributions is a taxonomy or "language" of camera motion primitives, designed in collaboration with cinematographers. We find, for example, that some primitives like "follow" (or `tracking`) require understanding scene content like moving subjects. We conduct a large-scale human study to quantify human annotation performance, revealing that domain expertise and tutorial-based training can significantly enhance accuracy. For example, a novice may confuse `zoom-in` (a change of intrinsics) with translating forward (a change of extrinsics), but can be trained to differentiate the two. Using CameraBench, we evaluate Structure-from-Motion (SfM) and Video-Language Models (VLMs), finding that SfM models struggle to capture semantic primitives that depend on scene content, while VLMs struggle to capture geometric primitives that require precise estimation of trajectories. We then fine-tune a generative VLM on CameraBench to achieve the best of both worlds and showcase its applications, including motion-augmented captioning, video question answering, and video-text retrieval. We hope our taxonomy, benchmark, and tutorials will drive future efforts towards the ultimate goal of understanding camera motions in any video. Project page: `https://linzhiqiu.github.io/papers/camerabench`

## 1   Introduction

> *We must perceive in order to move, but we must also move in order to perceive.*
> — J. J. Gibson, *The Ecological Approach to Visual Perception* [21]

Humans perceive the visual world through movement. Motion parallax [54], for instance, enables precise depth perception essential for navigating the physical world [20]. Similarly, camera motion is crucial for modern vision techniques that process videos of dynamic scenes. For example, Structure-from-Motion (SfM) [55, 64, 78] and Simultaneous Localization and Mapping (SLAM) [14, 18, 59] methods must first estimate camera motion (pose trajectory) to reconstruct the scenes in 4D. Likewise, without understanding camera motion, video-language models (VLMs) [61, 72, 75] would not fully perceive, reason about, or generate video dynamics.

**Human perception of camera motion.** Understanding camera motion comes naturally to humans because we intuitively grasp the "*invisible subject*" – the camera operator who shapes the video's viewpoint, framing, and narrative. For example, in a video tracking a child's first steps, one can sense a parent's joy through their handheld, shaky movement. Professional cinematographers and filmmakers even use camera motion as a tool [15, 58] to enhance visual storytelling and amplify the emotional impact of their shots. Hitchcock's iconic `dolly zoom` moves the camera forward while zooming out, maintaining the subject's framing while altering the background to create the impression

39th Conference on Neural Information Processing Systems (NeurIPS 2025) Track on Datasets and Benchmarks.

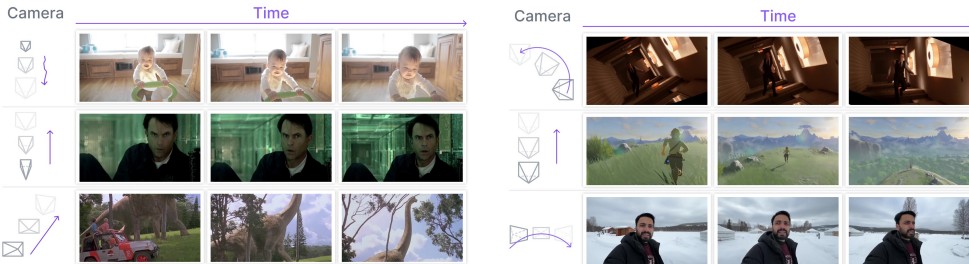

Figure 1: **Examples of camera movements.** We show videos with their camera trajectories: a `tracking shot` of a toddler (row 1, left), Hitchcock's `dolly zoom` effect (row 2, left), Spielberg's dramatic `pan` and `tilt` in *Jurassic Park* (row 3, left), Nolan's `roll` shot in *Inception* (row 1, right), a `pedestal-up` shot from *The Legend of Zelda* (row 2, right), and a selfie by an amateur photographer, `arcing` to showcase the scenery while centering themselves (row 3, right). Please watch the videos at our website.

of vertigo. In *Jurassic Park* (1993), Spielberg uses a slow `upward tilt` and `rightward pan` to evoke a sense of awe as the protagonists (and the audience) first see the dinosaurs. In *Inception* (2010), Nolan uses a camera `roll` to mirror shifting gravity, blurring the line of reality. Similarly, game developers use camera movement to enhance player immersion. In *Legend of Zelda: Breath of the Wild* (2017), a smooth `pedestal-up` shot transitions from the character's viewpoint to a breathtaking aerial view, hinting at the journey ahead. Even amateur photographers use camera motion as a tool; for example, selfie videos allow one to play the role of both the cinematographer and the subject. See Figure 1 for examples.

**Computational approaches to camera motion.** In contrast, classic computer vision methods learn camera motion from what is "visible" in the frame, relying on techniques like SfM and SLAM to estimate camera poses from video sequences. While these geometry-based approaches perform well on simple, static scenes, it is unclear how well they generalize to *dynamic, real-world videos* due to the difficulty of separating camera motion from scene dynamics [41, 66]. Moreover, these approaches do not capture the *high-level semantics* of camera motion [58], such as the intent behind a shot (e.g., tracking a subject or revealing a scene) or the context in which the motion occurs (e.g., handheld, gimbal-stabilized, or vehicle-mounted). On the other hand, recent multimodal vision systems like GPT-4o and Gemini [49, 52, 61] show strong human-like perceptual capabilities through large-scale training, yet their ability to understand camera motion remains largely untested. Inspired by these end-to-end approaches, we propose a *data-driven* framework for benchmarking and developing models that can perceive camera motion as humans do. However, this seemingly straightforward task poses challenges overlooked by prior work, as we detail next.

**Challenges and our approach.** We find major issues in widely-used datasets with camera motion annotations, such as MovieNet [30], AVE [1], and DREAM-1K [65]. First, many **lack a clear or correct specification of motion types**, often conflating fundamental concepts like translation with rotation or zoom. Second, these datasets often assign **contradictory labels** to the same video (e.g., labeling a video as both static and moving, which are mutually exclusive). Third, they **lack careful oversight**, resulting in significant annotation errors. To address these issues, we collaborate with professional cinematographers to develop a comprehensive taxonomy, a robust label-then-caption framework, and a training program backed by a large-scale human study to improve annotation quality. These efforts allow us to scale over 150K high-quality annotations across 3,381 videos.

**CameraBench.** We introduce **CameraBench** to benchmark and develop models for human-like understanding of camera motion, using our initial set of videos (each reviewed by at least one author during the quality control phase). Our comprehensive annotations, which include both labels and captions, allow us to evaluate models on a wide range of tasks, including binary classification of motion primitives, video-text retrieval, video captioning, and video question-answering (VQA). We evaluate a diverse set of 20 models, including discriminative [37, 38, 42, 52, 68] and generative VLMs [4, 36, 43, 49, 61, 77], and SfM/SLAM [41, 64, 66] methods. Although not all models can perform every task (e.g., SfM/SLAM cannot perform VQA tasks or reason about object-centric motion), we ensure fair comparisons by carefully designing the benchmarking protocol.

**Findings.** We find that classic SfM/SLAM methods [55] often fail to handle dynamic or low-parallax scenes (e.g, when the camera is stationary or only rotating), thus struggling with even classifying basic motion primitives (e.g., "*Is the camera moving up or not?*"). We also observe that recent

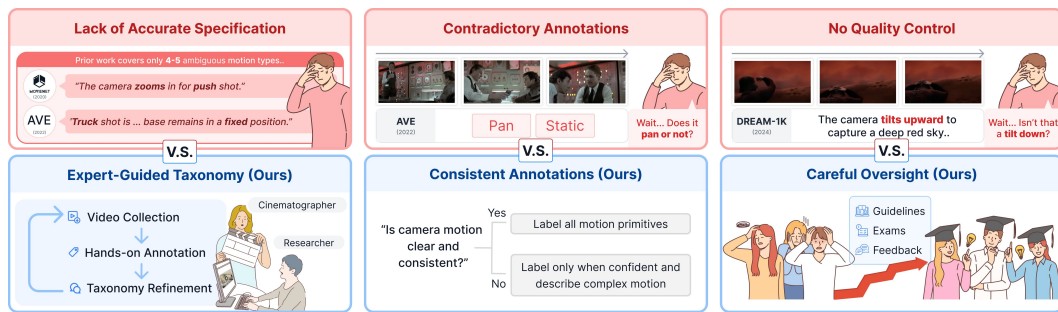

Figure 2: **Issues in previous camera motion datasets and our solutions.** Existing work contains critical flaws: (1) **Inaccurate specification**, e.g., MovieNet [30, 53] conflating translation with rotation or zoom. (2) **Contradictory annotations**, e.g., AVE [1] labels over 1,000 clips as both `static` (locked) and moving (including `pan` and `tilt`). (3) **No quality control**, even recent VLM benchmarks [5, 60, 65] contain major mistakes such as flipping motion direction. See Appendix A for analysis. Section 4 shows how we address them by working with professionals to design (1) a **taxonomy** via iterative refinement, (2) a reliable **annotation framework** for complex motion, and (3) a **training program** with expert oversight to improve data quality.

Table 1: **Comparison with prior human-annotated datasets.** We compare skill coverage, reference frame of motion, annotation format, and data quality. See Appendix A for a detailed report. A question mark indicates either confusion between translation, rotation, or zoom, or missing public information. CameraBench uniquely offers broader skill coverage, three reference frames (camera/object/ground), expert verification, manual shot segmentation, tutorial-based training, and rich labels and captions for benchmarking video-language models.

| Benchmark | Year | Data Access | #Label | Skill Coverage | | | | | Ref Frame | | | Expert Reviewed | Tutorial Trained | Multi Label | Motion Caption | Cut Method |
|---|---|---|---|---|---|---|---|---|---|---|---|---|---|---|---|---|
| | | | | Rot | Trans | Zoom | Arc | Track | Cam | Obj | Gnd | | | | | |
| MovieNet [30] | 2020 | ✓ | 4 | ? | ? | ? | ✗ | ✗ | ✓ | ✗ | ✗ | ✗ | ✗ | ✗ | ✗ | Auto |
| MovieShot [53] | 2021 | ✓ | 4 | ? | ? | ? | ✗ | ✗ | ✓ | ✗ | ✗ | ✗ | ✗ | ✗ | ✗ | Auto |
| AVE [1] | 2022 | ✓ | 5 | ? | ? | ? | ✗ | ✗ | ✓ | ✗ | ✗ | ✗ | ✗ | ✗ | ✗ | Auto |
| DREAM-1K [65] | 2024 | ✓ | ✗ | ✓ | ✓ | ✓ | ✗ | ✗ | ✓ | ✗ | ✗ | ✗ | ✗ | ✗ | ✓ | Auto |
| VDC [5] | 2024 | ✓ | ✗ | ✓ | ✓ | ✓ | ✗ | ✓ | ✓ | ✓ | ✗ | ✗ | ✗ | ✗ | ✓ | Auto |
| Cinematic2K [40] | 2024 | ✗ | 11 | ✓ | ✓ | ✓ | ✗ | ✓ | ✓ | ✓ | ✗ | ? | ? | ✗ | ? | Manual |
| VidComposition [60] | 2024 | ✓ | 7 | ? | ? | ? | ✗ | ✗ | ✓ | ✗ | ✗ | ✗ | ✗ | ✓ | ✗ | Auto |
| **CameraBench (Ours)** | 2025 | ✓ | 50 | ✓ | ✓ | ✓ | ✓ | ✓ | ✓ | ✓ | ✓ | ✓ | ✓ | ✓ | ✓ | Manual |

learning-based SfM/SLAM methods like MegaSAM [41, 66] handle dynamic scenes much better and outperform the classic COLMAP [55] by 1-2x. However, they may still confuse camera motion with object or scene motion in complex scenarios. We argue that our benchmark serves as a *reality check* for future SfM/SLAM methods, helping identify areas for improvement. On the other hand, we find that generative VLMs show promise in understanding camera motion, particularly in tasks requiring semantic reasoning (e.g., tracking shot). This motivates us to use our dataset to post-train VLMs for better camera motion understanding. With our small-scale yet high-quality fine-tuning data, we show that VLMs can achieve 1-2x improvements across both discriminative and generative tasks.

**Contributions.** We (1) introduce a taxonomy of camera motion primitives, developed in collaboration with domain experts; (2) design a robust annotation framework and training program to improve data quality; (3) collect a benchmark featuring real-world videos of dynamic scenes across diverse genres and motions; and (4) analyze the strengths and limitations of existing models to guide future research. We hope our data, taxonomy, and models can improve understanding of camera motions in any video.

## 2 Related Work

**Camera motion in vision datasets.** Existing datasets typically represent camera motion in three ways: **(1) Camera trajectory**. Per-frame camera poses provide a *geometric* description of motion, but obtaining ground-truth trajectories for real-world dynamic scenes is nearly impossible. For example, datasets [12, 29, 32, 45, 80] like RealEstate10K rely on multi-view geometry methods [55] to estimate *pseudo ground-truth* trajectories, and they are mostly limited to static scenes. To achieve more accurate trajectories, some datasets use simulators with camera control to generate synthetic videos [31, 56]. However, camera trajectories only offer a camera-centric view of motion, ignoring object and scene context. **(2) Motion labels**. Datasets with discrete labels often suffer from poor specification and cover only a limited set of motion categories. MovieNet [30, 53] defines only four types of movements and focus solely on movies. AVE [1] expands the taxonomy but confuses rotation

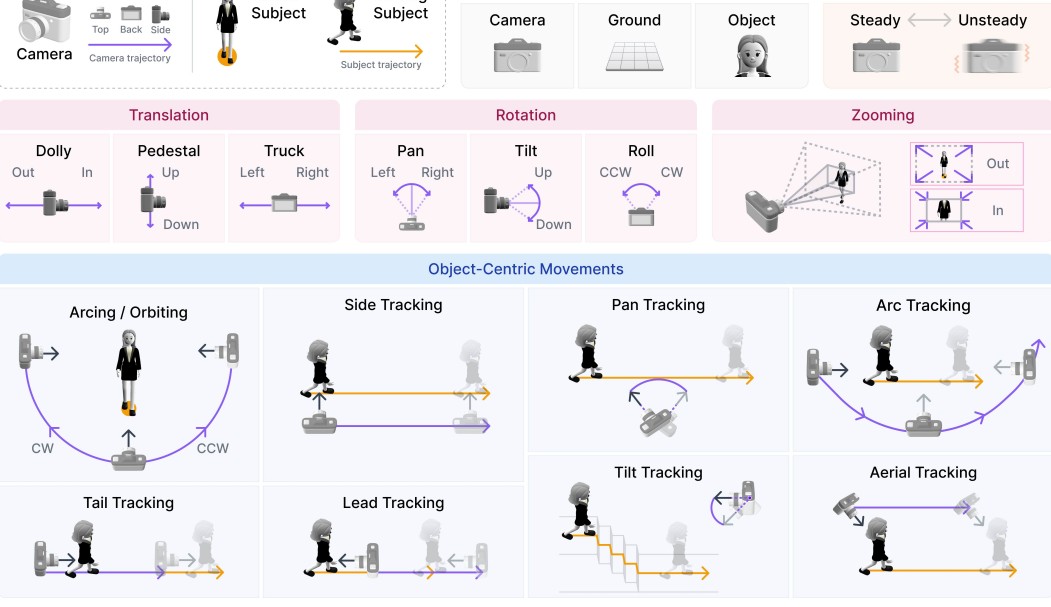

Figure 3: **Taxonomy of camera motion primitives.** Our taxonomy, developed in collaboration with cinematographers and vision researchers, is the first to comprehensively capture camera motion across object-, ground-, and camera-centric reference frames, using precise cinematography terms [15] to eliminate ambiguity. It covers camera steadiness, translation, rotation, intrinsic changes, and common object-centric movements, all detailed in this paper. We refine the taxonomy iteratively over three months by annotating real-world videos and incorporating feedback from researchers and cinematographers to ensure both accuracy and completeness.

as translation (e.g., grouping `pan` and `truck`) and intrinsic as extrinsic change (e.g., grouping `dolly` and `zoom`). We also find that AVE contains contradictory annotations, such as videos labeled as both "`static`" and "`pan`". Recent datasets [40] add object-centric motion labels like `tracking` shot but force videos into a single label, failing to capture co-occurring motions. **(3) Motion descriptions**. Recent video-language models [28, 40, 65] leverage human-collected motion descriptions, but their datasets, taxonomies, or annotation guidelines are either not open-source or undocumented. Lastly, we note that existing datasets that involve camera motion often have limited coverage of videos, featuring only static scenes [80], narrow domains (e.g., only movies [1, 30]), or unedited footage [23].

**Camera motion in generative models.** Our study is partly inspired by the growing interest in incorporating camera movement into video generative models. For instance, text-to-video generation models [70, 73] often learn camera control using synthetic camera movements, or are trained and evaluated on largely static scenes with SfM-estimated camera trajectories [2, 3, 10, 25, 33, 39, 47, 56, 69, 71, 72, 72, 79, 81]. Yet, it remains unclear whether SfM can reconstruct accurate trajectories for real-world or synthetic videos. While there is a large body of work analyzing the robustness of camera motion estimation using sensitivity analysis [11, 13, 19], these methods typically assume access to ground-truth 2D point correspondences, which are difficult to obtain in in-the-wild video sequences. More recently, models like MovieGen [51] and Skyreels [6] train in-house classifiers to augment captions with camera motion labels, while Goku [9] uses a captioner [75] to generate motion descriptions. However, none of these works have open-sourced their datasets.

## 3 Camera Motion Requires Clear Specification and Expert Oversight

We analyze seven previous datasets that claim to cover camera motion and identify critical issues that limit their usefulness. We summarize these issues, analyze why they arise, and outline our solutions.

**Key issues in prior datasets.** Many existing datasets suffer from one or more critical flaws. (1) They lack a clear or correct specification of motion. For example, MovieNet [30] incorrectly defines forward translation (`dolly-in`) as a `zoom`, conflating physical camera movement with intrinsic lens change. (2) Their annotation frameworks are often inconsistent [1], leading to contradictory labels such as assigning both `static` (locked) and `pan` to the same video. (3) They lack expert verification

and quality control. For instance, even recent test benchmarks [5, 60, 65] for video-language models contain over 50% errors when describing camera motion, e.g., hallucinating `tilt-down` as `tilt-up`. We provide interactive web viewers in the supplement to visualize these errors.

**Why these issues arise.** While humans can intuitively perceive camera motion, converting that perception into data annotations is far from trivial. First, motion can be ambiguous without a specified **reference frame**. For example, people might describe a bird's-eye-view camera moving "forward" along its optical axis as moving "downward", because it descends toward the ground. In general, humans tend to describe camera motion based on the scene or object context, such as saying "*The camera is following the subject*" in a tracking shot, while the camera actually leads the subject by moving backward (row 1, left of Figure 1). Many **camera movement terms are also misunderstood**. Amateurs often confuse `zoom-out` (intrinsic lens change) with `dolly-out` (extrinsic camera movement). Finally, while prior work often treats camera motion as a classification task [30, 51], **internet videos may contain complex motion patterns**. For example, a drone camera might smoothly move forward before abruptly reversing direction mid-flight, making it unreasonable to classify as either `dolly-in` or `dolly-out`.

**Our solution.** These challenges suggest that camera motion is harder to annotate than previously assumed and requires both accurate definitions and careful oversight (see Figure 2). This motivates us to work with professional cinematographers, who use precise terminology to describe motion when planning shots and communicating intent to directors and crew [58]. Our collaborators include film school students and professionals with over 10 years of experience from the US and China. Together, we develop a comprehensive taxonomy, a robust annotation framework, and an annotator training program, described next.

# 4 Taxonomy Design, Annotation Framework, and Training Program

We first introduce our taxonomy and annotation framework, then present a large-scale human study used to design a structured training program that significantly improves annotator performance.

**Iterating on the taxonomy with hands-on annotation.** We work closely with cinematographers, who use established terminology to describe how the camera moves to frame subjects, reveal scenes, and guide viewer perspective [15, 17, 58]. Our team takes a hands-on, iterative approach: over several months, we annotate real-world videos, hold weekly discussions to resolve disagreements, and refine label definitions by adding missing terms and clarifying edge cases. To capture diverse camera motion patterns, we source videos from platforms like YouTube across a wide range of **genres** (e.g., nature, film, advertisements, news, video games, abstract art, selfies, sports, tutorials, drone footage, studio productions, performance shows, screen recordings, vlogs, anime, motion graphics), **types** (2D, 2.5D, 3D, synthetic, real), **perspectives** (e.g., first-person, third-person), **devices** (e.g., smartphones, dashcams, GoPros, steadicams, fisheyes), and **post-production effects** (e.g., overlays, framings, mixed reality). We adhere to YouTube Standard licenses for all videos. Unlike prior datasets [1] that rely on automatic shot segmentation [57], we *manually* segment each video into single, continuous shots for accurate annotation. See Appendix B for detailed statistics.

**Taxonomy overview.** After reaching perfect consensus on an initial set of ~800 videos, our team finalizes a taxonomy of **over 50 motion primitives** (where prior work [1, 30] defines only 4 to 5). Due to space constraints, we present an overview in Figure 3, show example annotations in Figure 5, and refer readers to Appendix F for detailed definitions:

- **Motion type.** The camera motion is nonexistent (`no`), clear and consistent (`simple`), subtle (`minor`), or ambiguous/conflicting (`complex`).
- **Steadiness.** The camera remains still (`static`) or exhibits different levels of shakiness (`no shaking, minimal shaking, unsteady, very unsteady`).
- **Translation.** The camera physically moves forward or backward (`dolly`), up or down (`pedestal`), or to the right or left (`truck`).
- **Rotation.** The camera rotates along its own axis to the right or left (`pan`), up or down (`tilt`), or clockwise or counterclockwise (`roll`).
- **Intrinsic change.** The camera adjusts its focal length to zoom in or out (`zoom`).
- **Object-centric movements.** The camera orbits around a subject (or the frame center) in a circular path (`arc`), or tracks a moving subject from behind (`tail-tracking`), the front (`lead-tracking`),

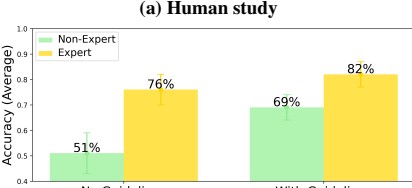

**(a) Human study**

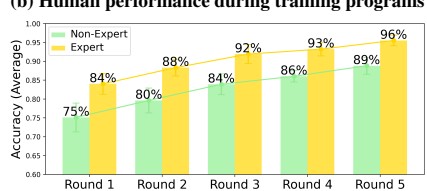

**(b) Human performance during training programs**

Figure 4: **Human study and training program.** We hire ∼100 participants from diverse backgrounds, including non-expert with limited knowledge about camera movements and experts from the filmmaking industry with hands-on cinematography experience. Figure **(a)** shows the average accuracy of both groups in selecting motion primitives on 30 videos, where experts clearly outperform non-experts. In addition, around 80% of participants who review our *multimodal* guidelines (including textual definitions, video examples, and edge cases) significantly outperform the remaining 20% who only see *textual* definitions. Figure **(b)** shows that extended practice with detailed error feedback boosts accuracy for all participants. We hire only those who complete all five rounds (with 30 videos each) to annotate our dataset.

the side (`side-tracking`), from an aerial view (`aerial-tracking`), or using other motions (`tilt-/pan-/arc-tracking`). We also consider whether the camera moves or zooms to make the subject appear `larger` or `smaller` within the frame.

- **Others.** We include the speed of camera movement (`slow/regular/fast`), cinematic effects (`dolly-zoom/motion-blur`), and scene movement (`static/mostly-static/dynamic`).

**Comments on the taxonomy.** We also specify the **motion direction** for the above primitives (`in/out/up/down/right/left/CW/CCW`). Humans tend to interpret camera translation relative to the ground due to a natural bias toward gravity: in Figure 5 (row 1, left), the camera moves forward (`dolly-in`) while pointing directly at the ground in a bird's-eye-view. Yet, most humans describe it as moving downward (`pedestal-down`). Appendix D explains how we resolve this ambiguity using two questionnaires to separately label camera translation in ground-centric and camera-centric frames. Finally, some primitives like steadiness and speed are inherently perceptual. To reduce subjectivity, we include reference videos in our labeling policy to improve annotator agreement. For model evaluation, we do not use these labels directly and instead focus on unambiguous questions (e.g., whether the camera shakes or not, rather than how much it shakes).

**Annotation framework.** A common approach to annotating camera motion is to treat each aspect as a classification task [1, 30], e.g., "*Does the camera pan right or left?*" with options like "`pan-right`", "`pan-left`", or "`no-pan`." However, real-world videos often contain conflicting or ambiguous motions, making direct classification unreliable. While recent work directly describes camera motion using natural language [40, 65], we find this approach error-prone. For instance, annotators often miss translation when rotation dominates the video. This challenge is amplified in our setup, as we intentionally source diverse videos that span single, consistent motions (e.g., `dolly-in`), compound motions (e.g., `dolly-in + zoom-out`), ambiguous motions (e.g., subtle movement or lack of depth), and sequential motions (e.g., `tilt-up` followed by `tilt-down`). To address these challenges, we adopt a "**label-then-caption**" approach to robustly annotate complex camera motion. First, annotators determine whether the camera motion is **clear and consistent**. If so, they classify each aspect directly. If motion is **ambiguous or conflicting**, they only answer when confident, leaving others as "*I am not sure*." These unanswered questions are excluded from the final dataset. Next, we ask annotators to provide a natural language description to capture conflicting movements (e.g., "*The camera first pans left, then right*") or uncertain cases (e.g., "*A 2D cartoon without depth cues to determine actual camera movement*"). To better capture how camera motion impacts visual storytelling, we encourage annotators to describe why the camera moves in a particular way, e.g., revealing the scene and following the subject.

**Human study for quality annotation.** We use our expert-annotated videos to conduct a human study using LabelBox under an educational license. We recruit over 100 participants via crowdsourcing platforms, university and film school boards, and professional studios. These participants come from diverse backgrounds – half with cinematography experience (professional cinematographers and film school students) and half without (graphic/UI/UX designers, freelancers, and college students from fields like literature and computer science). Initially, 20 participants annotate 30 videos based on our taxonomy definitions. Figure 4-(a) shows that expert participants with cinematography experience outperform non-experts by more than 15% in accuracy.

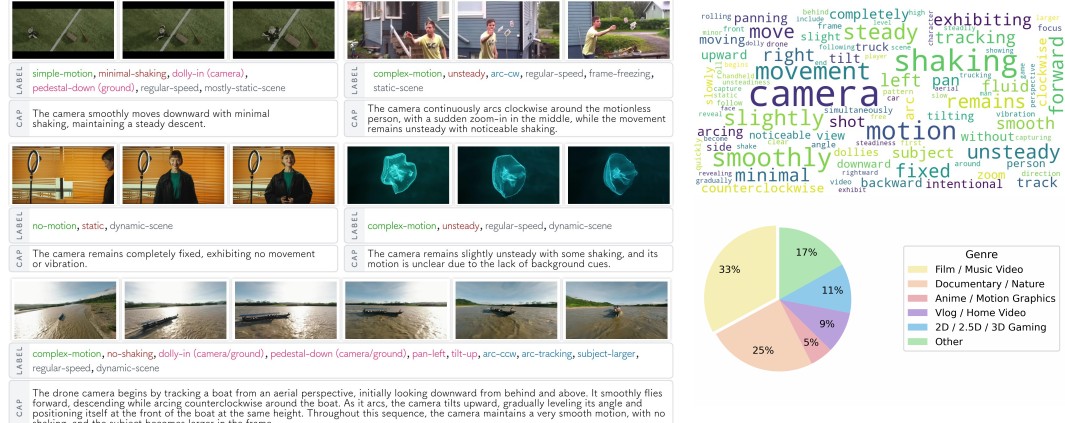

Figure 5: **Example annotations.** Our videos (**left**) are annotated with binary labels for ∼50 camera motion primitives from our taxonomy, along with language descriptions capturing key motion aspects. We visualize the caption word cloud on the **top-right** and a pie chart of video genres on the **bottom-right**. Note that the `other` genre includes more tags such as dashcam, drone, selfie, ads, mixed media, animals, art, sports, lectures, screen recordings, and etc. See our website for videos.

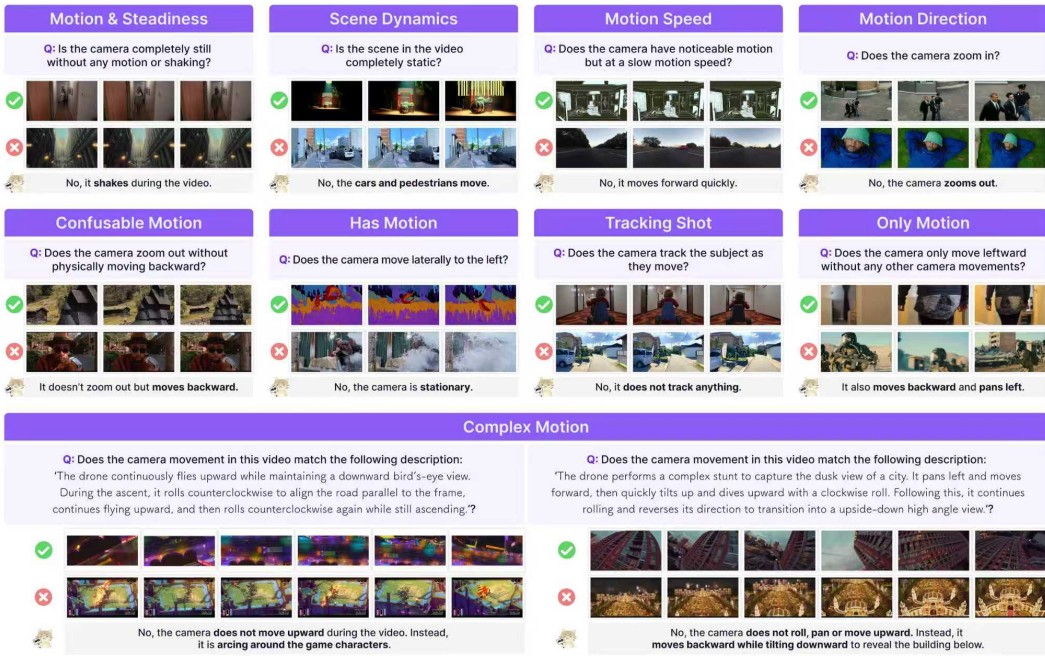

Figure 6: **VQA examples of CameraBench.** We evaluate 9 challenging camera motion understanding skills (with 81 sub-tasks detailed in Appendix G). Each question is paired with a positive video (answer: "Yes") and a negative video (answer: "No"), ensuring a vision-centric benchmark that cannot be solved blindly [22, 35, 43].

**Training program for improving annotation performance.** Non-experts often struggle with confusable motions, such as **rotation vs. translation** or **extrinsic vs. intrinsic changes**, due to a limited understanding of parallax effects [54]. To address this, we prepare training materials with detailed textual guidelines, positive/negative video examples, and edge cases. Figure 4-(a) shows that our tutorials benefit not just non-experts – even cinematographers finding the examples helpful. Next, incoming annotators attend lectures given by the authors and complete five more rounds of exams (30 videos each). After each exam, we send a detailed feedback report to help them correct misunderstandings. Figure 4-(b) shows that extended practice further improves performance by 10-15% as participants better align with our policy. We hire only those who successfully complete all training and continuously monitor their performance through random audits. For any disagreements, we hold feedback sessions and revise annotations to reach consensus. See Appendix D for details on

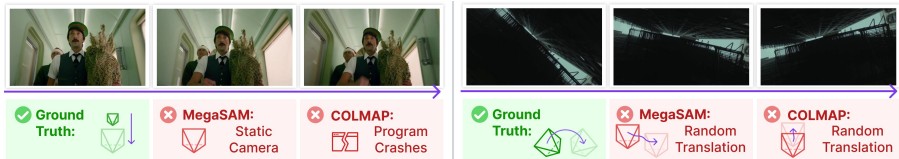

Figure 7: **Failures of SfM/SLAM. Left:** a `lead-tracking` shot where the camera moves backward (relative to the ground) as the subject walks forward. Since the subject's framing remains unchanged and the background lacks distinct textures, MegaSAM [41] fails to detect camera translation and COLMAP [55] crashes. **Right:** a `roll` shot in a low-parallax scene where both methods do not converge and output nonexistent translation.

Table 2: **Binary classification on motion primitives defined in the camera-centric frame.** We report Average Precision per primitive. We find that (1) recent SfM/SLAM methods like MegaSAM [41] significantly outperform COLMAP [55], but all methods remain far from solving this task with ∼50% AP. (2) Generative VLMs clearly outperform discriminative ones. Motivated by this, we fine-tune Qwen2.5-VL [4] on a separate training set of ∼1400 videos (no overlap with the test set). We show that simple SFT (highlighted in green) significantly boosts performance by 1-2x, making it match the SOTA MegaSAM in overall AP. We **bold** the best and underline the second-best results; finetuned VLMs are ranked separately.

| Model | Translation (Dolly/Pedestal/Truck) | | | | | | Zooming | | Rotation (Pan/Tilt/Roll) | | | | | | Static | Avg |
|---|---|---|---|---|---|---|---|---|---|---|---|---|---|---|---|---|
| | In | Out | Up | Down | Right | Left | In | Out | Right | Left | Up | Down | CW | CCW | | |
| Random Chance | 29.3 | 9.7 | 6.7 | 8.6 | 15.8 | 11.5 | 11.1 | 10.2 | 15.0 | 15.4 | 12.7 | 7.7 | 8.9 | 10.2 | 9.7 | 12.2 |
| *SfM/SLAM* | | | | | | | | | | | | | | | | |
| COLMAP | 36.2 | 13.1 | 11.9 | 19.7 | 34.1 | 30.0 | 13.9 | 14.2 | 43.9 | 46.4 | 28.3 | 19.1 | 42.1 | 48.7 | 7.5 | 27.3 |
| VGGSFM | 56.6 | 28.9 | 28.7 | 38.2 | 48.9 | 35.3 | 21.7 | 17.3 | 60.9 | 58.7 | 46.6 | 43.3 | 61.4 | 55.5 | 16.7 | 41.3 |
| DUSt3R | 70.3 | 37.3 | 41.7 | 30.2 | 41.5 | 35.6 | 18.2 | 24.6 | 59.4 | 63.8 | 32.9 | 27.3 | 61.0 | 57.9 | 42.6 | 43.0 |
| MASt3R | 65.4 | 34.3 | 35.1 | 59.6 | 43.7 | 38.1 | 42.2 | 46.6 | 66.6 | 58.0 | 63.2 | 40.3 | 50.4 | 53.5 | 45.6 | 49.5 |
| CUT3R | 88.0 | 65.5 | 38.7 | 54.6 | 42.5 | 36.5 | 15.9 | 21.3 | 59.1 | 65.0 | 65.0 | 47.5 | 60.7 | 66.2 | 37.6 | 50.9 |
| MegaSAM | 87.0 | 58.3 | 43.0 | 48.4 | 59.1 | 58.0 | 11.1 | 10.2 | 77.9 | 82.4 | 75.6 | 57.7 | 67.4 | 76.9 | 60.1 | 58.2 |
| *CLIPScore* | | | | | | | | | | | | | | | | |
| UMT-B16-CLIP | 27.0 | 10.4 | 9.0 | 20.0 | 19.4 | 11.8 | 11.8 | 9.9 | 11.9 | 13.5 | 13.1 | 8.4 | 18.8 | 15.6 | 10.0 | 14.0 |
| UMT-L16-CLIP | 27.2 | 9.8 | 12.3 | 10.8 | 18.5 | 11.5 | 17.5 | 8.9 | 16.0 | 17.4 | 21.9 | 8.3 | 7.3 | 10.0 | 13.0 | 14.0 |
| LanguageBind-CLIP | 32.7 | 13.2 | 7.8 | 11.2 | 14.2 | 11.7 | 14.4 | 9.4 | 20.1 | 16.4 | 14.1 | 8.5 | 13.8 | 9.5 | 10.9 | 13.9 |
| LanguageBindV1.5-CLIP | 33.6 | 14.5 | 11.0 | 10.3 | 15.0 | 11.8 | 14.2 | 10.1 | 19.9 | 16.7 | 16.1 | 9.2 | 17.6 | 10.2 | 10.4 | 14.7 |
| InternVideo2-S2-CLIP | 41.7 | 9.4 | 5.8 | 9.7 | 15.0 | 12.0 | 15.0 | 9.9 | 20.6 | 18.8 | 14.7 | 9.1 | 8.3 | 10.8 | 11.4 | 14.2 |
| *ITMScore* | | | | | | | | | | | | | | | | |
| UMT-B16-ITM | 31.7 | 11.5 | 11.4 | 14.3 | 16.6 | 12.8 | 12.3 | 9.2 | 15.1 | 16.9 | 16.2 | 10.0 | 14.2 | 12.1 | 8.9 | 14.2 |
| UMT-L16-ITM | 40.6 | 10.6 | 8.5 | 17.6 | 21.9 | 23.6 | 12.4 | 9.8 | 21.3 | 33.2 | 31.0 | 11.2 | 13.5 | 12.3 | 9.4 | 18.4 |
| InternVideo2-S2-ITM | 52.4 | 12.6 | 10.5 | 14.7 | 15.8 | 19.7 | 21.1 | 16.7 | 29.4 | 29.1 | 24.5 | 18.4 | 17.2 | 13.4 | 14.0 | 20.6 |
| *VQAScore* | | | | | | | | | | | | | | | | |
| LLaVA-OneVision-7B | 46.8 | 13.5 | 12.6 | 16.9 | 23.7 | 20.2 | 10.7 | 14.4 | 33.5 | 33.6 | 16.9 | 31.4 | 19.3 | 20.8 | 18.8 | 22.2 |
| LLaVA-Video-7B | 54.7 | 15.2 | 16.5 | 19.3 | 27.1 | 23.6 | 16.2 | 16.9 | 33.6 | 36.8 | 26.9 | 37.2 | 16.1 | 21.7 | 22.1 | 25.6 |
| InternVideo2-Chat-8B | 69.9 | 18.5 | 19.3 | 17.6 | 17.9 | 23.4 | 12.2 | 10.4 | 22.6 | 22.7 | 17.2 | 22.8 | 19.6 | 16.4 | 20.2 | 22.0 |
| Tarsier-Recap-7B | 59.7 | 15.1 | 25.7 | 23.7 | 28.8 | 21.5 | 14.4 | 15.0 | 22.8 | 27.3 | 24.6 | 21.6 | 15.2 | 18.7 | 30.7 | 21.0 |
| InternLMXComposer2.5-7B | 49.0 | 10.6 | 11.4 | 10.4 | 14.6 | 10.6 | 11.8 | 16.5 | 14.3 | 13.9 | 14.7 | 17.5 | 11.7 | 18.1 | 21.8 | 16.5 |
| InternVL2.5-8B | 67.9 | 12.9 | 28.1 | 25.9 | 23.4 | 23.2 | 18.6 | 32.1 | 37.4 | 30.9 | 37.6 | 36.9 | 11.5 | 25.3 | 23.4 | 29.5 |
| InternVL2.5-26B | 63.6 | 11.8 | 21.1 | 23.6 | 27.2 | 19.4 | 21.8 | 31.6 | 42.5 | 38.3 | 44.9 | 43.6 | 14.3 | 18.2 | 25.1 | 29.8 |
| mPLUG-Owl3-7B | 47.6 | 12.9 | 13.9 | 16.9 | 17.3 | 18.5 | 12.9 | 10.6 | 31.4 | 26.6 | 26.1 | 37.0 | 10.4 | 12.2 | 17.8 | 20.8 |
| GPT-4o | 66.3 | 29.2 | 21.1 | 38.2 | 38.0 | 21.9 | 41.7 | 39.3 | 44.7 | 42.1 | 43.6 | 35.5 | 24.0 | 28.7 | 32.0 | 36.4 |
| InternVL3-8B | 61.2 | 15.5 | 18.8 | 29.0 | 30.5 | 27.3 | 29.5 | 28.1 | 41.6 | 49.3 | 42.0 | 36.5 | 21.3 | 22.3 | 20.1 | 31.5 |
| InternVL3-78B | 72.0 | 18.2 | 19.6 | 32.5 | 33.8 | 29.4 | 26.4 | 33.4 | 47.2 | 53.5 | 47.8 | 40.3 | 27.6 | 25.0 | 22.6 | 36.8 |
| Qwen2.5-VL-7B | 56.0 | 14.9 | 18.7 | 30.5 | 34.5 | 27.6 | 29.8 | 43.4 | 62.7 | 66.7 | 54.5 | 34.1 | 18.8 | 24.2 | 19.8 | 35.8 |
| Qwen2.5-VL-32B | 57.6 | 16.4 | 20.2 | 32.1 | 36.0 | 29.2 | 31.4 | 45.0 | 64.3 | 68.2 | 56.0 | 35.6 | 20.2 | 25.7 | 21.2 | 37.3 |
| Qwen2.5-VL-72B | 58.0 | 16.8 | 20.6 | 32.5 | 36.4 | 29.5 | 31.7 | 45.4 | 64.7 | 68.6 | 56.4 | 36.0 | 20.6 | 26.1 | 21.6 | 37.7 |
| **Qwen2.5-VL-7B (Ours SFT)** | 83.9 | 38.6 | 27.8 | 47.8 | 67.9 | 50.0 | 54.5 | 75.8 | 79.2 | 83.8 | 76.3 | 67.6 | 32.3 | 41.0 | 73.6 | 60.0 |
| **Qwen2.5-VL-32B (Ours SFT)** | 85.6 | 40.1 | 29.3 | 49.4 | 69.6 | 51.5 | 56.0 | 77.3 | 80.7 | 85.4 | 77.9 | 69.2 | 33.9 | 42.7 | 75.4 | 61.6 |
| **Qwen2.5-VL-72B (Ours SFT)** | 86.8 | 41.3 | 30.5 | 50.6 | 70.7 | 52.6 | 57.1 | 78.5 | 81.9 | 86.6 | 79.1 | 70.4 | 35.0 | 43.8 | 76.6 | 62.8 |

this process. As of this writing, we have over 150K binary labels across 3,381 fully annotated videos.

# 5   CameraBench for Motion Understanding

We repurpose our motion primitive labels and captions for both **discriminative** (classification, retrieval) and **generative** (VQA, captioning) tasks.

**Baselines.** We evaluate a diverse set of **20 models**, including **6 SfM/SLAM** methods: COLMAP [55] and learning-based variants such as MegaSAM [41], CUT3R [66], and others [16, 64, 67]. We also report **3 discriminative VLMs** [38, 82] like InternVideo2 [68] and **11 generative VLMs** including Qwen2.5-VL [4], GPT-4o [49], and LLaVA-Video [77], among others [36, 61, 68, 75, 76].

**Classification of motion primitives.** We evaluate models on binary classification of motion primitives, restricted to those defined in the camera-centric frame to align with SfM/SLAM outputs. For SfM/SLAM, we compute the seven degrees of translation, rotation, and focal change from estimated camera extrinsics and intrinsics (if available) between the first and last frame. For discriminative VLMs, we use textual definitions of each primitive ("*The camera pans to the left.*") to compute matching scores. For generative VLMs, we compute VQAScore [44], i.e., the probability of "Yes" to a binary question ("*Does the camera pan to the left?*"). Appendix G details prompts for VLMs.

**Results.** Table 2 shows that (1) learning-based SfM/SLAM methods like MegaSAM significantly outperform COLMAP and set the state-of-the-art. Nonetheless, no methods fully solve this task, as the best overall AP remains ∼50%. Figure 7 shows failure cases, e.g., SfM/SLAM struggles with low-parallax (rotation only) scenes. (2) While weaker than SfM/SLAM, generative VLMs like GPT-4o show promising results, significantly outperforming discriminative VLMs. This motivates us to fine-tune Qwen2.5-VL using supervised fine-tuning (SFT) on a separate set of ∼1400 videos (with no overlap with the testset). Despite the small dataset size, our SFT model achieves ∼2x performance, matching that of MegaSAM. We note that certain motions like `roll` remain particularly challenging for VLMs, likely due to their long-tailed nature [50] in internet videos.

**Beyond camera-centric motion primitives.** We collect ∼10K VQA samples across 9 top-level skills and 81 sub-tasks. Crucially, these tasks go beyond camera-centric frame reasoning to evaluate more aspects such as object-centric motion, scene dynamics, steadiness, and more. Some tasks also require logical (e.g., verifying if *only one* motion type exists or if a motion is *absent*) and linguistic reasoning (e.g., checking if a motion description is accurate). We follow community best practices [22, 35], pairing each question with two videos with opposite answers so that models cannot answer blindly without seeing the video (see Figure 6).

**VQA results.** Table 3 shows that all open-source VQA models perform at or below chance on CameraBench. Nonetheless, our SFT model – fine-tuned on our small training set – achieves state-of-the-art results across all skills, especially the most challenging ones (e.g., Tracking Shot and Only Motion) that require object-centric and logical reasoning.

**Other tasks.** We summarize key findings: (1) **Captioning** (Figure 8). We prompt VLMs with "*Describe the camera movements in this video*". Our SFT model generates more accurate captions than state-of-the-art VLMs, both qualitatively and quantitatively, as measured by metrics like SPICE and LLM-as-a-Judge. (2) **Video-text retrieval** (Table 4). We use video pairs in CameraBench's VQA tasks to evaluate retrieval performance and show that generative VLMs (using the discriminative VQAScore [44]), outperform other baselines. (3) **Motion control in image-to-video generation** (Figure 17). While we focus on video understanding, we note that finetuning CogVideoX1.5-I2V [74] using CameraBench can potentially improve its camera motion control.

# 6   Conclusion

**Limitations.** Future work may explore post-training techniques beyond SFT [24, 42]; for example, optimizing preset prompts [46] could further improve VLM performance. We leave camera motion control in video generation as future work. Lastly, given the complementary strengths of SfM/SLAM and VLMs, integrating them could be promising for advancing video understanding.

**Conclusions.** We take the first step toward human-like camera motion understanding by introducing a taxonomy of motion primitives and a robust annotation framework, developed in collaboration with cinematographers. We implement a training program to transform laypeople into proficient annotators of camera movements. We curate a diverse benchmark to analyze existing models and

Table 3: **VQA evaluation.** We report both accuracy (**Acc**) and question accuracy (**Q-Acc**) [35] that scores a point only if *both* videos are answered correctly for a given question. We **bold** the best and underline the second-best results; finetuned models (highlighted in green) are ranked separately. While most VLMs perform at or below chance, our SFT model achieves the best overall performance.

| Model | Motion & Steadiness Acc | Q-Acc | Scene Dynamics Acc | Q-Acc | Motion Speed Acc | Q-Acc | Motion Direction Acc | Q-Acc | Confusable Motion Acc | Q-Acc | Has Motion Acc | Q-Acc | Shot Tracking Acc | Q-Acc | Only Motion Acc | Q-Acc | Complex Description Acc | Q-Acc | Avg Overall Acc | Q-Acc |
|---|---|---|---|---|---|---|---|---|---|---|---|---|---|---|---|---|---|---|---|---|
| Random Chance | 50.0 | 25.0 | 50.0 | 25.0 | 50.0 | 25.0 | 50.0 | 25.0 | 50.0 | 25.0 | 50.0 | 25.0 | 50.0 | 25.0 | 50.0 | 25.0 | 50.0 | 25.0 | 50.0 | 25.0 |
| mPLUG-Owl3-7B | 51.8 | 15.5 | 64.9 | 35.1 | 61.5 | 31.6 | 48.6 | 13.1 | 49.2 | 12.7 | 54.1 | 24.3 | 53.2 | 17.1 | 45.9 | 8.6 | 63.4 | 39.7 | 55.8 | 25.4 |
| LLaVA-Video-7B | 53.5 | 12.8 | **66.1** | **36.2** | 57.2 | 22.4 | 52.1 | 17.8 | 49.9 | 5.4 | 54.9 | 13.9 | 59.9 | 29.2 | 51.3 | 2.9 | 68.0 | **41.8** | 58.8 | 24.1 |
| LLaVA-OneVision-7B | 54.3 | 19.6 | 63.8 | 31.0 | 69.0 | **54.0** | 53.1 | 24.2 | **55.4** | 20.7 | 60.9 | 28.2 | **60.7** | **31.3** | 43.3 | 6.1 | 52.3 | 6.3 | 57.1 | 24.7 |
| InternVideo2-Chat-8B | 52.4 | 13.7 | 64.4 | 31.6 | 51.7 | 5.2 | 50.2 | 2.9 | 49.7 | 13.8 | 52.2 | 5.5 | 48.5 | 2.3 | 50.9 | 4.3 | 50.6 | 1.3 | 51.3 | 5.3 |
| Tarsier-Recap-7B | 51.8 | 12.3 | 62.8 | 29.2 | 50.5 | 4.8 | 49.8 | 2.5 | 49.0 | 12.5 | 51.5 | 5.0 | 47.8 | 2.0 | 50.2 | 3.8 | 49.8 | 1.0 | 50.6 | 4.8 |
| InternLMXComposer2.5-7B | 52.8 | 12.8 | 57.8 | 19.5 | 56.6 | 17.2 | 49.6 | 1.7 | 53.3 | 14.8 | 53.2 | 9.9 | 49.1 | 11.6 | 51.2 | 2.4 | 48.4 | 7.8 | 51.7 | 9.3 |
| InternVL2.5-8B | 54.4 | 14.9 | 59.8 | 23.0 | 57.5 | 31.6 | 51.3 | 12.8 | 49.7 | 0.0 | 58.1 | 22.5 | 55.2 | 14.1 | 50.0 | 0.0 | 50.0 | 0.0 | 54.5 | 16.7 |
| InternVL2.5-26B | 56.2 | 17.3 | 63.5 | 26.4 | 60.8 | 35.2 | 53.8 | 15.6 | 51.2 | 14.5 | 60.3 | 25.8 | 58.4 | 18.9 | 52.5 | 2.4 | 53.6 | 3.8 | 57.2 | 19.8 |
| InternVL3-8B | 54.4 | 14.9 | 59.8 | 23.0 | 57.5 | 31.6 | 51.3 | 12.8 | 49.7 | 0.0 | 58.1 | 22.5 | 55.2 | 14.1 | 50.0 | 0.0 | 50.0 | 0.0 | 54.5 | 16.7 |
| InternVL3-78B | 56.2 | 17.3 | 63.5 | 26.4 | 60.8 | 35.2 | 53.8 | 15.6 | 51.2 | 14.5 | 60.3 | 25.8 | 58.4 | 18.9 | 52.5 | 2.4 | 53.6 | 3.8 | 57.2 | 19.8 |
| Qwen2.5-VL-7B | 55.7 | 20.8 | 60.6 | 24.1 | 69.0 | 40.2 | 55.8 | 23.5 | 51.7 | 20.7 | 60.4 | 28.1 | 57.2 | 25.2 | 48.4 | 11.5 | 66.6 | 38.8 | 58.4 | 25.9 |
| Qwen2.5-VL-32B | 57.2 | 22.0 | 62.1 | 25.4 | 70.5 | 41.5 | 57.3 | 24.7 | 53.2 | 21.9 | 61.9 | 29.3 | 58.7 | 26.3 | 49.8 | 12.7 | 68.1 | 40.0 | 59.9 | 27.1 |
| Qwen2.5-VL-72B | 57.7 | 22.4 | 62.1 | 25.8 | **71.0** | 41.4 | 57.8 | 25.1 | 53.2 | **22.2** | 62.4 | 29.7 | 59.2 | 26.3 | 50.4 | 13.1 | **68.6** | **40.4** | **60.3** | 27.4 |
| GPT-4o | 55.8 | 27.0 | 52.6 | 10.3 | 61.2 | 32.2 | **58.1** | **32.8** | 53.3 | 20.4 | **64.1** | **36.2** | 51.7 | 20.2 | 42.1 | 8.5 | 61.9 | 32.7 | 59.0 | **29.8** |
| Gemini-2-Flash | 53.6 | 25.2 | 46.8 | 2.9 | 56.6 | 29.3 | 44.5 | 17.2 | 41.1 | 8.8 | 46.5 | 20.5 | 46.5 | 24.1 | 39.2 | 5.1 | 63.8 | 37.4 | 51.8 | 24.9 |
| Gemini-2.5-Pro | **58.2** | **28.7** | 51.3 | 11.6 | 60.1 | 34.5 | 48.9 | 21.4 | 45.7 | 13.2 | 52.3 | 25.8 | 49.7 | 26.9 | 42.8 | **15.3** | 64.5 | 39.1 | 54.7 | 28.2 |
| **Qwen2.5-VL-7B (Ours SFT)** | 72.2 | 48.0 | 75.6 | 53.4 | 81.6 | 63.2 | 70.3 | 46.3 | 54.7 | 13.3 | 75.2 | 54.9 | 75.9 | 52.0 | 59.9 | 21.2 | 77.0 | 55.0 | 71.4 | 45.3 |
| **Qwen2.5-VL-32B (Ours SFT)** | 74.0 | 49.5 | 77.4 | **55.0** | 83.5 | 64.8 | 72.2 | 47.8 | 56.4 | 14.7 | **77.1** | 56.4 | 77.7 | 53.4 | 61.6 | 22.6 | 78.7 | 56.5 | 73.2 | 46.8 |
| **Qwen2.5-VL-72B (Ours SFT)** | **74.5** | **49.9** | **77.9** | 55.0 | **83.5** | **65.2** | **72.7** | **48.2** | **57.0** | 14.7 | 77.1 | **56.8** | **78.2** | **53.8** | **62.1** | **23.0** | **79.2** | **56.9** | **73.6** | **47.1** |

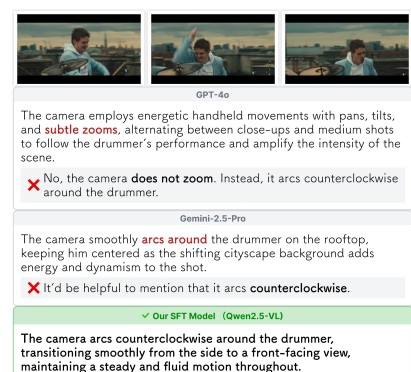

| Model | Caption Generation SPICE | ROUGE-L | BLEU-2 | METEOR | LLM-Judge |
|---|---|---|---|---|---|
| mPLUG-Owl3-7B | 0.22 | 0.20 | 0.08 | 0.19 | 0.08 |
| LLaVA-Video-7B | 0.23 | **0.23** | **0.12** | 0.19 | 0.09 |
| LLaVA-OneVision-7B | 0.22 | 0.21 | 0.10 | 0.20 | 0.09 |
| InternVideo2-Chat-8B | 0.22 | 0.21 | 0.11 | 0.19 | 0.13 |
| Tarsier-Recap-7B | 0.23 | 0.22 | 0.11 | 0.20 | 0.14 |
| InternLMXComposer2.5-7B | 0.21 | 0.19 | 0.08 | 0.19 | 0.10 |
| InternVL2.5-8B | 0.20 | 0.10 | 0.04 | 0.21 | 0.08 |
| InternVL2.5-26B | 0.23 | 0.20 | 0.09 | 0.23 | 0.11 |
| InternVL3-8B | 0.20 | 0.15 | 0.05 | 0.17 | 0.08 |
| InternVL3-78B | 0.18 | 0.16 | 0.06 | 0.18 | 0.07 |
| Qwen2.5-VL-7B | 0.18 | 0.12 | 0.05 | 0.28 | 0.16 |
| Qwen2.5-VL-32B | 0.24 | 0.17 | 0.08 | 0.29 | 0.18 |
| Qwen2.5-VL-72B | **0.25** | 0.19 | 0.10 | **0.30** | **0.19** |
| GPT-4o | 0.20 | 0.16 | 0.06 | 0.25 | 0.10 |
| Gemini-2-Flash | 0.24 | 0.21 | 0.10 | 0.22 | 0.07 |
| Gemini-2.5-Pro | 0.20 | 0.15 | 0.06 | 0.27 | 0.14 |
| **Qwen2.5-VL-7B (Ours SFT)** | 0.48 | 0.45 | 0.31 | 0.44 | 0.20 |
| **Qwen2.5-VL-32B (Ours SFT)** | 0.52 | 0.50 | 0.35 | 0.46 | 0.22 |
| **Qwen2.5-VL-72B (Ours SFT)** | **0.54** | **0.53** | **0.38** | **0.47** | **0.23** |

Figure 8: **Camera motion captioning. Left:** Example camera motion descriptions generated by our SFT model vs. GPT-4o and Gemini-2.5-Pro (see more in Figure 15 and Figure 16). **Right:** Automated evaluation of camera motion captions. We use both standard metrics (e.g., SPICE) and LLM-as-a-judge. For the latter, we prompt GPT-4o with: "*Reference caption: "{reference}" Candidate caption: "{candidate}" Does the candidate caption match the reference caption? Answer Yes or No.*" We then report the average confidence score P(Yes) [44].

suggest directions for future improvement. Lastly, we show that our high-quality dataset can be used to fine-tune VLMs for improved camera motion understanding.

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

# Towards Understanding Camera Motions in Any Video

## Supplementary Material

### *Outline*

Below is the outline of the supplement:

- **Section A** provides a detailed error analysis of prior datasets.
- **Section B** shows more statistics and examples of CameraBench.
- **Section C** details the annotation framework.
- **Section D** details our guidelines, training program, and quality control pipeline.
- **Section E** details the experimental setup and provides additional results.
- **Section F** details our label taxonomy.
- **Section G** details the 9 top-level skills and 81 sub-tasks in CameraBench.

## A    Error Analysis of Prior Datasets

We document key issues in seven widely-used datasets and benchmarks that claim to cover camera motion. Because many errors are best understood visually, we encourage readers to explore the original videos and our expert annotations via the interactive HTML reports linked below.

**Detailed issues in prior datasets.** Many existing datasets suffer from one or more of the following problems:

(1) **Lack of clear or correct specification.** For example, MovieNet [30] and MovieShot [53] incorrectly define forward translation (`dolly-in`) as a `zoom`, quoting "*the camera zooms in for a push shot*", thereby conflating physical camera movement with intrinsic lens change. AVE [1] conflates rotation with translation by grouping `pan` and `truck` into the same category, and defines this group as "*when the camera is moving horizontally while its base remains in a fixed position*", which is blatantly incorrect from a cinematographer's perspective. Other testing benchmarks [5, 40, 60, 65] do not provide any taxonomy or definition for each label at all.

(2) **Inconsistent annotation frameworks.** AVE [1] labels over 500 video clips as both `static` (locked) and `pan`, which are mutually exclusive. None of the prior datasets provide clear guidelines for annotating conflicting or compound motions, such as `pan-left` followed by `pan-right`, or `truck-left` combined with `zoom-in`.

(3) **No expert verification.** Even recent test benchmarks such as VidComposition [60], DREAM-1K [65], and VDC [5], which claim to include high-quality human-written captions or QA pairs, contain significant errors when describing or reasoning about camera motion. Common issues include mislabeling motion type, incorrect direction, or omitting motion entirely.

(4) **Additional issues.** These include missing common motion types (e.g., arc, tracking shots), unclear reference frames (e.g., "move down" without specifying whether it's ground-relative or camera-relative), no handling of shot transitions (treating multiple disjoint clips as a single shot), and narrow domain coverage (e.g., film-only datasets).

**Detailed reports.** Below we highlight representative issues in recent datasets, some with links to interactive reports for further inspection:

- **MovieNet and MovieShot (2020 and 2021)**: These two datasets are the earliest with human-annotated camera motion labels, but they only include four coarse types: `zoom-in` (for both forward movement and zooming in), `zoom-out` (for backward movement or zooming out), `static` (no motion), and `pans and tilts` (for any lateral movement or rotation). This specification is clearly inaccurate and incomplete, prompting follow-up work like AVE [1] to address these limitations.

- **AVE (2022)** (link to our interactive web viewer): AVE [1] defines five motion types: `pan/truck`, `tilt/pedestal`, `locked`, `zoom/dolly`, and `handheld`. This is a clear improvement over earlier datasets by separating pans and tilts and considering steadiness. However, it still conflates translation with rotation and zoom. Our expert team reviews the shot motion labels of 50 randomly sampled clips from AVE, and find that the error rate exceeds the accuracy, with more than half containing incorrect or contradictory annotations. In addition, over 1,000 clips are labeled as both `static` (locked) and motion types such as `pan` or `tilt`. We believe this results from a lack of clear labeling guidelines for handling inconsistent motions, as well as the absence of expert review during crowd-sourced annotation.

- **VDC (2024)** (link to our interactive web viewer): We review 20 randomly sampled captions from the VDC benchmark, which claims human review and serves as ground-truth for the CVPR'25 LOVE detailed video captioning challenge. We provide a detailed critique of their camera descriptions (with video IDs) in our interactive web viewer. Most captions omit both motion type and direction, and frequently hallucinate non-existent motion such as pans and zooms. In this sample, 60% of the captions fail to correctly describe camera motion.

- **DREAM-1K (2024)** (link to our interactive web viewer): DREAM-1K was first introduced in Tarsier [65] to evaluate detailed video captioning. While the paper claims to cover camera motion, the benchmark includes only sparse and often vague motion descriptions. Only few captions mention camera movement, and those that do frequently contain factual errors – such as hallucinating motion direction (e.g., `pan-left` as `pan-right`) or conflating translation with rotation (e.g., describing `tilt-down` as moving downward). In a random sample of 30 videos, only ∼30% of the motion-related descriptions were accurate.

- **VidComposition (2024)** (link to our interactive web viewer): We first note that this video QA benchmark [60] contains many uncut videos – each composed of multiple disjoint clips with distinct camera motions – making it unclear which clip the question refers to. After retrieving the ground-truth answers from the official evaluation server, we are still unable to determine their labeling policy. Our best guess is that a motion label is applied if any clip in the video shows the motion; otherwise, most of their answers would be clearly incorrect. Following this assumption, our expert team conducts a random audit of 20 QA pairs from VidComposition and found that over 55% were inaccurate. Several questions had multiple valid answers, and others had wrong answers (e.g., a `truck-left` shot was mis-labeled as `pan-left`). Also, although their paper appendix suggests this benchmark asks about tracking motion, we are unable to find such questions. By an exhaustive search of their dataset, we are only able to find seven motion types: `pan-up`, `pan-down`, `pan-left`, `pan-right`, `zoom-in`, `zoom-out`, and `static`. Lastly, although this benchmark provides a caption for each video, the captions completely omit any mention of camera movement.

- **Cinematic2K (2024)**: Because this dataset [40] is not open-sourced, we can only gather information from their technical report, which claims to have 11 motion types: `pan-left`, `pan-right`, `tilt-up`, `tilt-down`, `dolly-in`, `dolly-out`, `tracking-shot`, `zoom-in`, `zoom-out`, `rack-focus`, and `still`.

We invite readers to explore these examples and videos to better understand the challenges of annotating camera motion and the need for rigorous specification and expert oversight.

## B  CameraBench Details

**Dataset statistics.** CameraBench consists of 3,381 video clips with an average duration of 5.7 seconds and a frame rate of 29.4 FPS. The training split includes 1,402 videos. Using the same set of skills and tasks (detailed in Appendix G), we generate 230K video-QA pairs and 1,402 video-caption pairs for training.

**Word clouds.** Figure 9 shows the word cloud of our collected camera motion descriptions and metadata such as shot compositions, genres, points of views, and capturing devices.

**More examples.** Figure 10 presents more annotation examples from our dataset.

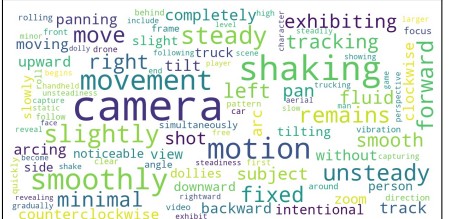

Figure 9: **Word clouds** of our camera motion captions (**left**) and metadata (**right**), including genres, types, shot compositions, point of views, capturing devices, and post-production effects.

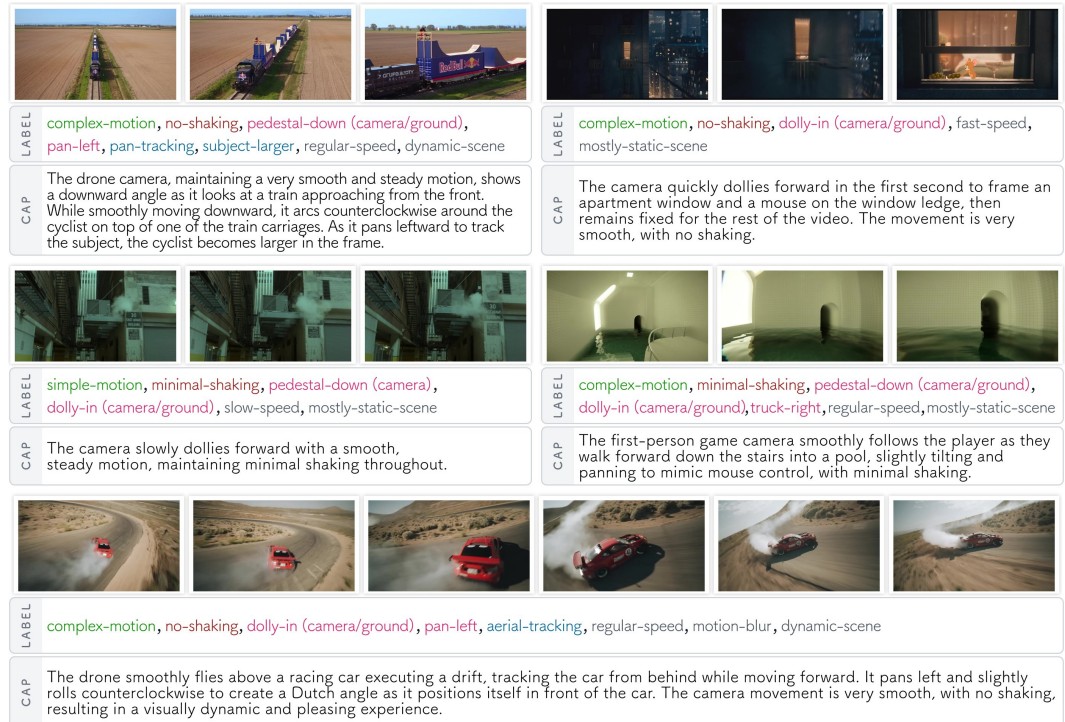

Figure 10: **More annotation examples from our dataset.**

## C   Annotation Framework

**Framework.** We design our annotation framework to ensure precision and efficiency by preventing contradictory labels and eliminating redundant work. We detail how we annotate the ∼50 motion primitives and descriptions below. Given a video, we first ask:

- **Is there camera motion?** First, check if the video has any camera motion (including small movements like handshakes). If yes, select the motion steadiness; otherwise, select `static` and then stops.

- **Is the motion clear and consistent?** If there is camera motion, choose `simple` for clear and consistent motion, `complex` for ambiguous or conflicting motion, or `minor` for small, barely noticeable motion.

Next, if the camera motion is `simple`, all motion primitives must be labeled comprehensively; otherwise, they are treated as negative samples (e.g., a `simple-motion` video not labeled as `pan-right` or `pan-left` is automatically assigned to `no-pan`). For `complex-motion` or `minor-motion` videos, annotators only select clearly identifiable, unambiguous primitives (e.g., consistent and non-conflicting motion). For example, if a camera first performs `dolly-in` and then `dolly-out`, the video is labeled as `complex`, with none of `dolly-in`, `dolly-out`, or `no-dolly` assigned. In these scenarios, annotators provide a description explaining the complex motion patterns. If the motion is

too intricate to fully describe, they should focus on what is clear and noticeable or simply state the reason for the camera movement (e.g., "*a handheld shot tracking a subject*" or "`a first-person camera following a person's perspective as they look around`"). For 2D anime or cartoons, we ask annotators to select `complex-motion` (except for only zooming motion), as these videos lack depth cues to determine actual camera movement. Note that for camera translation, we ask annotators to label and describe movement relative to the ground, as this aligns with most people's intuition. We then use a separate questionnaire to re-label videos with camera-centric translation primitives, including `dolly` and `pedestal`.

**Annotation interface.** Figure 11 shows the annotation interface we use, and Figure 12 lists example questions in our annotation framework. This interface allows annotators to watch the video and revise their answers as many times as needed before submission, as it is common to adjust previous labels based on later questions.

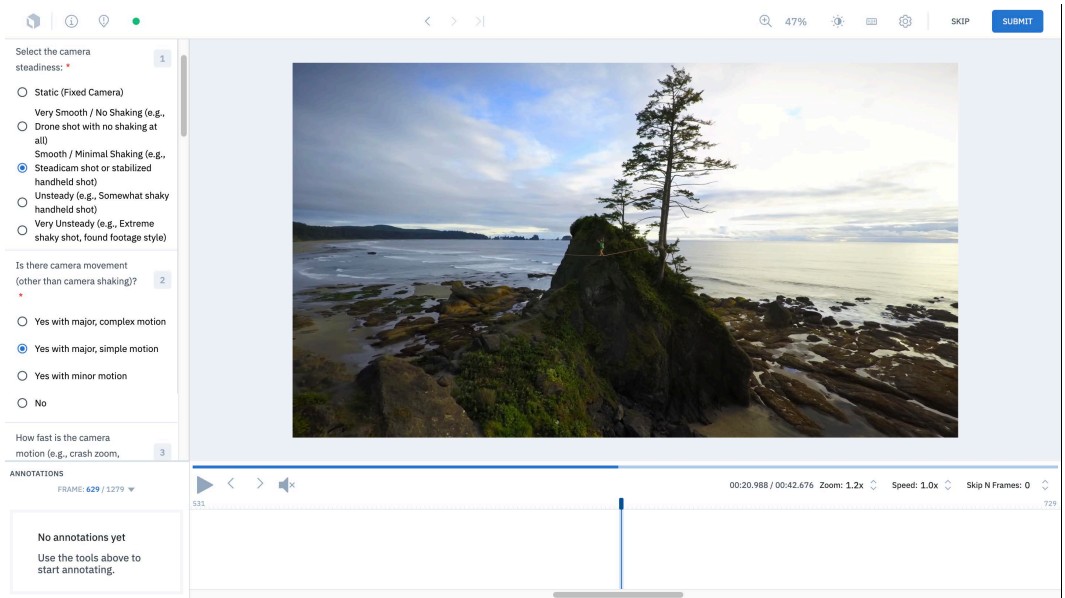

Figure 11: **Annotation interface** based on LabelBox.

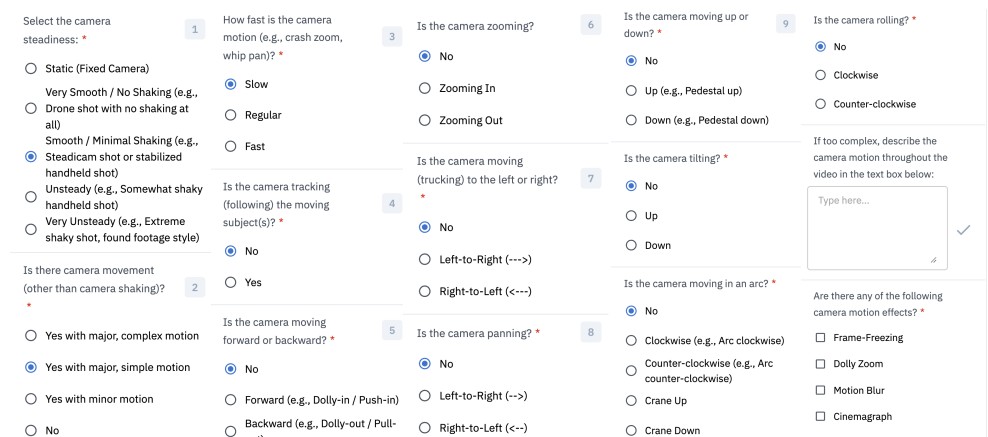

Figure 12: **Example questions** in our annotation framework.

# D    Training Program and Quality Control

**Tutorials.** To help participants familiarize themselves with camera movements and align with our labeling policy, we provide a tutorial with clear guidelines, textual definitions, video examples, and complex edge cases. Figure 13 shows a few random pages from our guidelines.

**Caption guidelines.** Labeling `complex-motion` videos can be challenging when movements are conflicting, sequential, occur at different speeds, or lack sufficient background or depth cues. To improve clarity in such complex scenarios, we ask annotators to provide descriptions that include (1) the **purpose** of the movement (if clear), such as following a subject, revealing a scene, or enhancing immersion; (2) the **major camera motions**, such as panning, arcing, or zooming, and whether the movement is steady or shaky. We ask annotators to provide details when the motions are sequential and easy to perceive. If the motion is highly intricate or fragmented, we ask them to write a high-level summary instead.

**Caption quality.** For motion descriptions, we ask annotators to focus on the following three criteria: (1) **clearness:** *Does the description clearly convey the intended information?* (2) **conciseness:** *Is the description expressed in as few words as possible without losing clarity?* (3) **grammar and fluency**: *Does the text sound natural and free of errors?* Annotators are encouraged to use LLMs like ChatGPT to polish their initial description (e.g., for grammar refinement). The suggested prompt is: `Please help me polish my text to make it clear, concise, and grammatically correct. Maintain the intended meaning and tone while improving readability. Avoid using overly complex or fancy words unless necessary. If the text includes specific details, ensure they remain intact. Additionally, make sure the polished version flows naturally and is easy to understand.`

**Training program.** Before annotating the main dataset, participants undergo five rounds of training, each with 30 videos. After each round, they receive a detailed PDF report (Figure 14) showing their accuracy and a comparison with the ground truth, helping them review and refine their responses. If participants still have doubts, the authors of this paper offer direct guidance. After five rounds, their performance typically improves by 15–20%.

**Quality control pipeline.** We hire only annotators who successfully complete all training. Each annotator is then assigned a specific role to ensure annotation accuracy and consistency:

1. **Labeler:** Each video is independently labeled by two labelers.
2. **Reviewer:** Reviewers check for consensus and resolve label disagreements.

Beyond these roles, the authors of this paper conducted an additional review of all videos, correcting inaccurate labels and refining motion descriptions to ensure clarity and accuracy.

# E    Experimental Setup and Results

**More video captioning examples.** Figure 15 and Figure 16 compare our SFT model with other VLMs on more videos.

**Video-text retrieval results.** Table 4 and Table 5 show **Text Score**, **Video Score**, and **Group Score** on all video-text retrieval tasks.

**Motion control for image-to-video generation.** While our main focus is on video understanding, we conduct a preliminary experiment by fine-tuning CogVideoX-1.5 (5B) [74] to generate video from a single input image and a caption describing camera motion. Using the CameraBench training split, we fine-tune the model and evaluate on randomly selected test samples (Figure 17).Compared to the original CogVideoX, the fine-tuned model shows improved control over camera motion such as dolly, zoom, and arc. We plan to explore video generation and its evaluation more deeply in future work. We plan to further explore video generation and evaluation in future work [7, 8, 27, 48, 63].

**VLM details.** For discriminative VLMs, we adapt their official codebases to compute CLIPScore [26, 52] and ITMScore [37] for video-text matching scores. For generative VLMs, we also adapt their official codebases but implement the logic to calculate VQAScore [34, 44] for discriminative scoring. While GPT-4o provides a logprob API for computing VQAScore, Gemini-2/2.5 disables its logprob API during this work. We note that almost all VLMs utilize uniform frame sampling; however, the

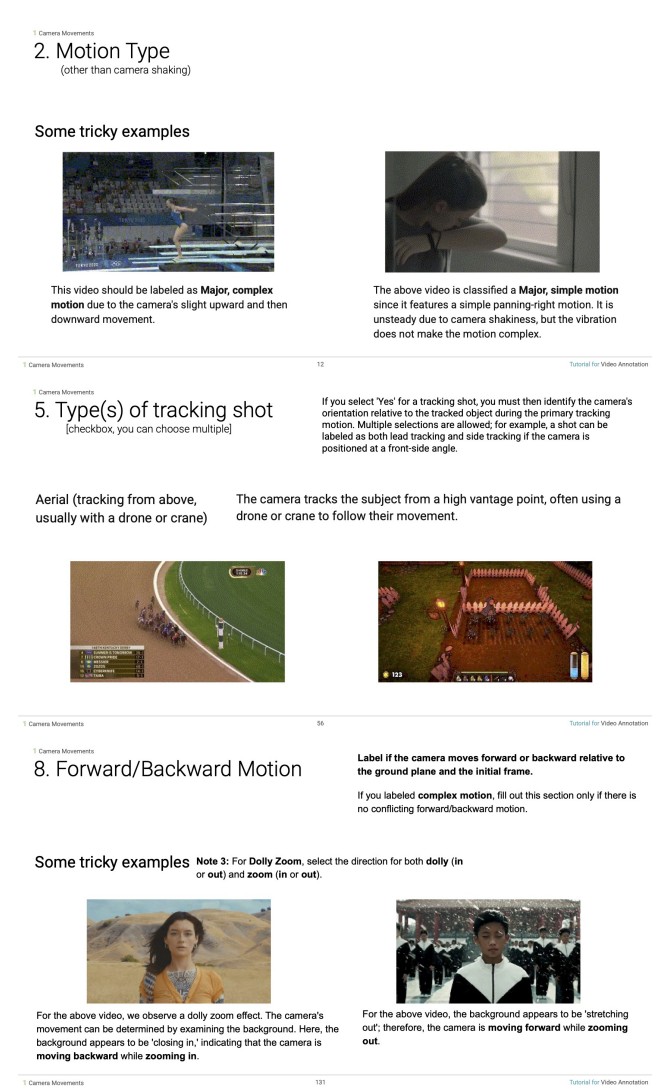

Figure 13: **Example guidelines from our tutorial.**

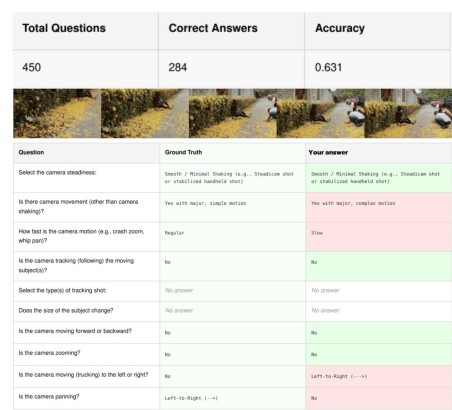

Figure 14: **Examples of our PDF feedback to participants.** Wrong answers are colored in red.

number of frames used varies across models. To ensure optimal performance on our dataset, we use

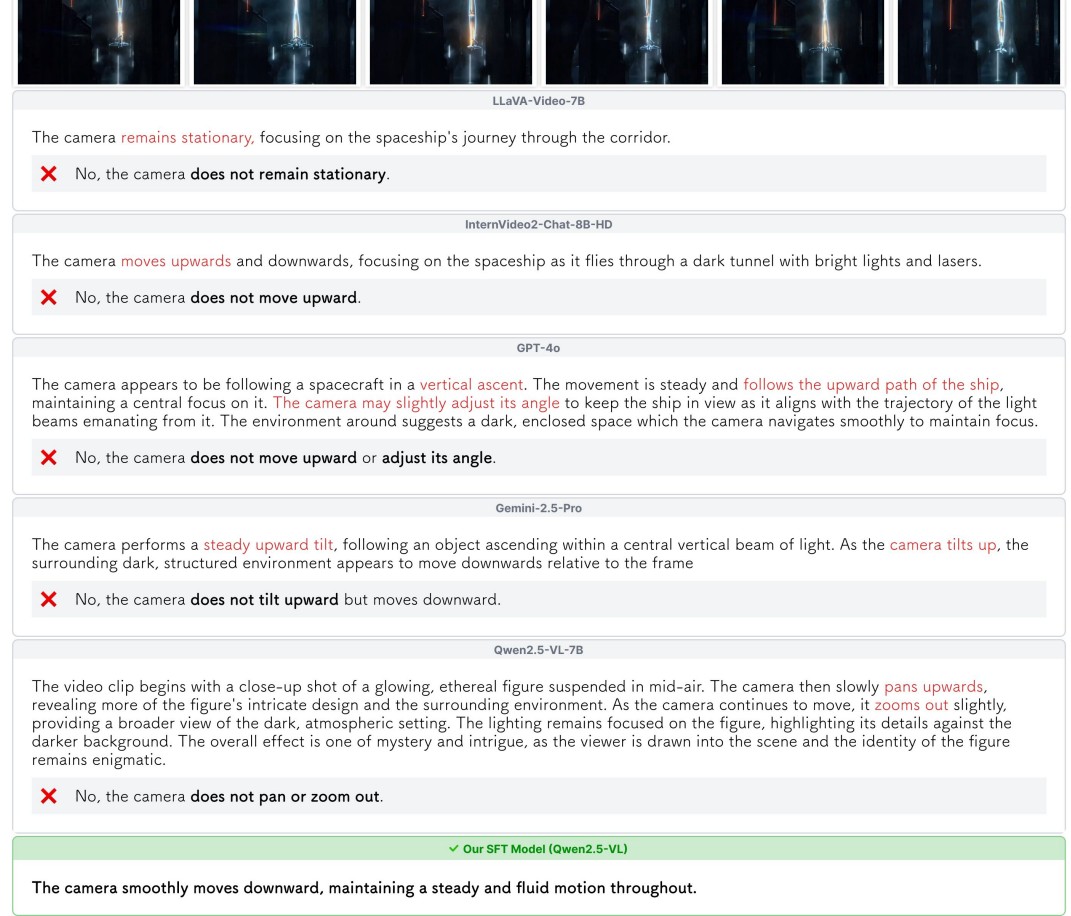

Figure 15: **Comparing motion descriptions for different VLMs (example 1 of 2).**

the recommended number of frames for each model. We set the number of frames sampled to 4 for GPT-4o. Notably, some models deviate from simple uniform sampling. Gemini-2/2.5 [61] processes video file inputs directly, with its frame sampling procedure hidden from the user. We also note that Qwen2.5-VL [4] uses frames-per-second (FPS) sampling. Unlike uniform sampling, FPS sampling ensures a consistent number of frames per second of video.

We use a separate training set of ∼1,400 videos (with no overlap with the test set) to fine-tune Qwen2.5-VL [4] using the official supervised fine-tuning code. Our main results are based on full fine-tuning. For full fine-tuning, we adopt DeepSpeed ZeRO-3 while freezing the vision tower and multi-modal projector. Training was done for 5 epochs. The learning rate for the 7B model was 2.0e-5 and 1.0e-5 for the 32B and 72B models, with cosine scheduling and a warmup ratio of 0.05. We use a multinode setup with 3 8-GPU nodes of NVIDIA H-100 GPUs. Hyperparameter details for the best runs are shown in Table 6, Table 7, and Table 8. We ablate the number of frames sampled per second (FPS) using Qwen-2.5-7B finetuned on training set using different FPS rates on the binary classification tasks, and observe a consistent performance boost with higher FPS (e.g., 8) outperforming lower FPS (e.g., 2) across the board. Results are shown in Table 9. As such, we stick with 8 FPS for our SFT models. To finetune our model, we make use of the LLaMA-Factory codebase. All settings are the same for all 3 model sizes except for the learning rates. For comparison, we also run LoRA fine-tuning (rank 64) with a slightly higher learning rate of 2e-4 on the 7B model, which we find to be optimal. We found full fine-tuning to outperform LoRA fine-tuning after 5 epochs.

**SfM/SLAM details.** We benchmark six classic and learning-based SfM and SLAM methods. For COLMAP [55], we use the default parameters for feature extraction, matching, and mapping but replace exhaustive matching with sequential matching using a window size of 10 to balance accuracy and speed. Due to COLMAP's sensitivity to initialization, we also evaluate VGGSfM [64], which

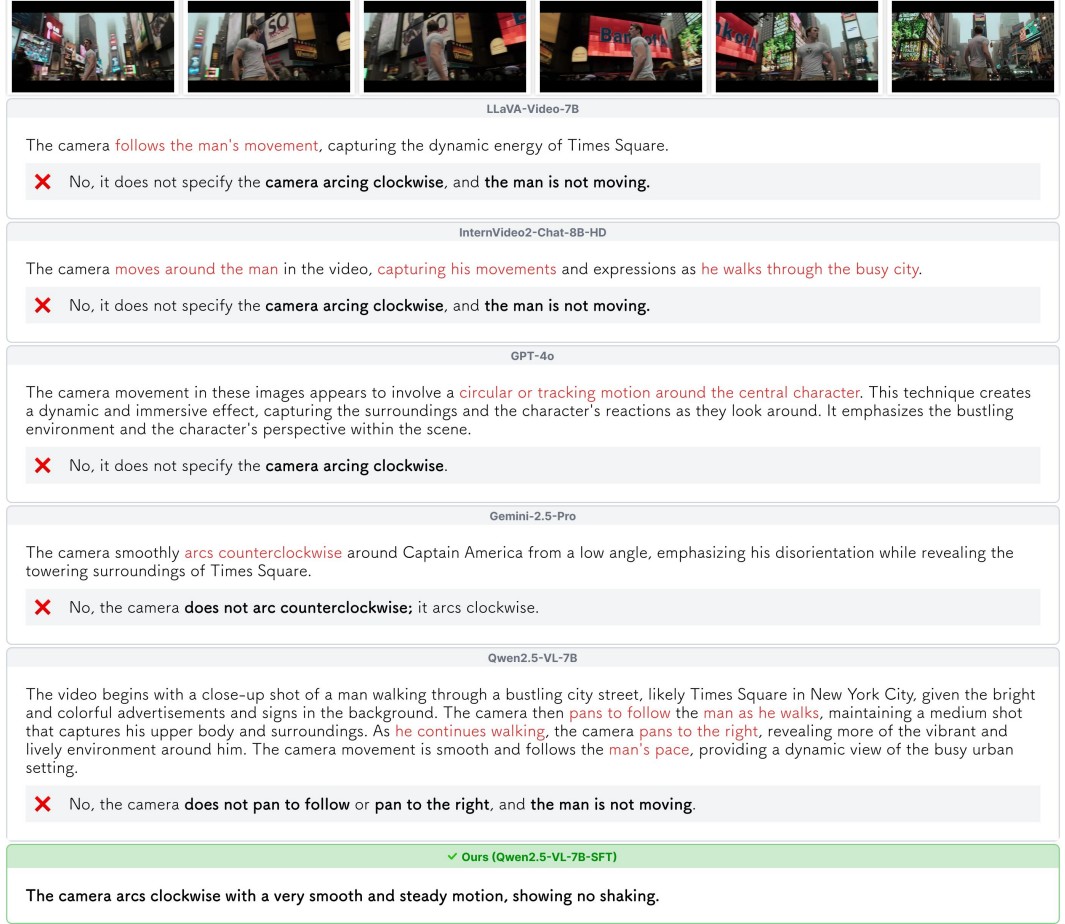

Figure 16: **Comparing motion descriptions for different VLMs (example 2 of 2).**

incorporates a learning-based front-end for feature extraction and matching, along with a learnable camera and point initializer for improved convergence. We observe that VGGSfM converges quickly and therefore use exhaustive matching for this method while keeping its default hyperparameters. Additionally, we evaluate DUST3R [67], MAST3R [16], and CUT3R [66], which propose a unified paradigm for solving 3D tasks using pointmap prediction. To benchmark MAST3R efficiently, we replace its default exhaustive pair optimization strategy with a more efficient sparse optimization method to prevent out-of-memory (OOM) errors. For all these methods, we resize the longer side of images to 512 and utilize their 512-size checkpoints, aligning with the official evaluation procedures. Finally, we evaluate MegaSAM [41], a recently released method designed for 4D reconstruction in dynamic videos. We use its default parameters but skip the final causalSAM step, as it optimizes only the depth rather than the points. To convert the camera poses obtained from SfM and SLAM methods into motion primitive scores, we use a straightforward approach based on the normalized relative pose between the first and last frames of the trajectory. We calculate the normalization factor by calculating the average distance of all reconstructed 3D points to the origin. The motion scores are derived as follows: translation scores are directly taken from the relative translation values along the three axes, while rotation scores are computed from the relative rotation along the roll, pitch, and yaw axes. We convert all axes to align with OpenCV's axis convention to ensure consistency. Lastly, the zoom score is determined by calculating the ratio of the focal lengths between the first and last frames. For CUT3R and MegaSAM, we use a video sampling strategy of max(30FPS, 200 frames) to ensure continuous motion. In contrast, for COLMAP, DUST3R, and Mast3R, we sample at 1 FPS to enable efficient inference and avoid OOM errors. We further ablate MegaSAM's performance at 2, 4, and 8 FPS and observe only minimal differences compared to the default sampling strategy in Table 9.

Table 4: **Evaluation on video-text retrieval.** We compare CLIPScore, ITMScore, and VQAScore models on skill-based and caption-based video-text retrieval tasks, measured by Text, Video, and Group scores as defined in [35, 62]. **Skill-based** task refers to evaluating on all 8 skills except for **Complex Description**. **Caption-based** task refers to evaluating on the **Complex Description** skill. We show that repurposing generative VLMs (especially our SFT model) for discriminative scoring using VQAScore sets the state-of-the-art.

| Model | Skill-based Task | | | Caption-based Task | | |
|---|---|---|---|---|---|---|
| | Text | Image | Group | Text | Image | Group |
| Random Chance | 25.0 | 25.0 | 16.6 | 25.0 | 25.0 | 16.6 |
| *CLIPScore* | 21.6 | 5.8 | 3.5 | 44.0 | 26.7 | 19.8 |
| UMT-B16 | 26.8 | 4.1 | 2.8 | 46.0 | 19.0 | 13.0 |
| UMT-L16 | 23.7 | 4.4 | 2.6 | 39.5 | 17.3 | 11.1 |
| LanguageBind | 24.0 | 9.7 | 6.2 | 53.6 | 39.6 | 33.2 |
| LanguageBindV1.5 | 24.1 | 8.3 | 5.4 | 55.9 | 38.7 | 33.0 |
| InternVideo2-S2 | 9.3 | 2.3 | 0.7 | 25.0 | 18.9 | 8.6 |
| *ITMScore* | 17.6 | 9.5 | 4.3 | 42.7 | 37.2 | 25.3 |
| UMT-B16 | 14.7 | 9.1 | 3.9 | 30.6 | 33.0 | 18.7 |
| UMT-L16 | 19.9 | 10.7 | 5.0 | 45.2 | 37.0 | 26.2 |
| InternVideo2-S2 | 18.2 | 8.7 | 4.1 | 52.3 | 41.7 | 31.0 |
| *VQAScore* | 28.3 | 39.7 | 20.5 | 54.2 | 53.0 | 39.0 |
| mPLUG-Owl3-7B | 26.2 | 38.4 | 19.6 | 57.6 | 52.8 | 42.7 |
| LLaVA-OneVision-7B | 24.3 | 39.7 | 18.8 | 56.4 | 53.0 | 40.9 |
| LLaVA-Video-7B | 17.8 | 40.9 | 13.3 | 53.5 | 50.7 | 37.2 |
| InternVideo2-Chat-8B | 21.4 | 18.0 | 8.0 | 41.2 | 26.3 | 16.1 |
| Tarsier-Recap-2 | 35.1 | 23.1 | 15.4 | 43.4 | 30.4 | 22.6 |
| InternLMXComposer-2.5-7B | 14.3 | 33.0 | 9.8 | 40.4 | 54.2 | 29.5 |
| InternVL-2.5-8B | 22.0 | 43.9 | 17.5 | 55.8 | 51.4 | 38.7 |
| InternVL-2.5-26B | 22.1 | 45.1 | 18.7 | 57.4 | 54.2 | 39.1 |
| InternVL-3-8B | 31.9 | **46.0** | 25.0 | 60.2 | 57.3 | 45.8 |
| InternVL-3-78B | 35.7 | 44.6 | 26.8 | 63.4 | 60.5 | 48.2 |
| Qwen2.5-VL-7B | 35.0 | 40.8 | 24.2 | 65.5 | 63.0 | 51.8 |
| Qwen2.5-VL-32B | 41.4 | 42.7 | 29.5 | 65.6 | 67.7 | 53.0 |
| Qwen2.5-VL-72B | **43.8** | 44.5 | **32.1** | 67.8 | 69.2 | 56.4 |
| GPT-4o | 38.3 | 42.4 | 25.8 | 39.9 | 40.3 | 31.6 |
| **Qwen2.5-VL-7B (SFT)** | 44.6 | 59.1 | 42.7 | 83.4 | 85.2 | 76.7 |
| **Qwen2.5-VL-32B (SFT)** | 45.8 | 61.2 | 43.9 | 83.5 | 86.2 | 77.6 |
| **Qwen2.5-VL-72B (SFT)** | 46.3 | 62.2 | 44.4 | 83.5 | 86.7 | 78.1 |

## F    Full Taxonomy

We provide the full taxonomy below:

**Motion type.** The camera motion is nonexistent (`no`), clear and consistent (`simple`), subtle (`minor`), or ambiguous/conflicting (`complex`). Refer to Table 10 for details.

**Steadiness.** Steadiness affects visual clarity and motion perception in video analysis. While professional cinematography favors stability, intentional shake adds stylistic effects, like in handheld footage. We select if the camera remains still (`static`) or exhibits different levels of shakiness (`no shaking, minimal shaking, unsteady, very unsteady`). Refer to Table 11 for details.

**Translation.** The camera physically moves forward or backward (`dolly`), up or down (`pedestal`), or to the right or left (`truck`). Refer to Table 12 for definitions. Note that for camera translation, the choice of reference frame is crucial for consistent annotation. We define two reference frames: (1) The **camera-centric** reference frame defines motion relative to the camera's own coordinate system, where translations like forward and backward follow the camera's initial orientation. While widely used in existing datasets, it can sometimes be unintuitive for human perception. (2) In contrast, the **ground-centric** reference frame defines motion relative to the "world" coordinate system, typically the ground plane. To ensure we label direction consistently in the ground-centric reference frame, we define forward motion (`dolly-in`) in a bird's-eye view (looking directly downward at the ground) as moving "north" or toward the top of the frame, and backward motion (`dolly-out`) as moving "south" or toward the bottom. Similarly, in a worm's-eye view (looking directly upward at the sky), forward motion is defined as moving "south" (toward the bottom of the frame), and backward motion as moving "north" (toward the top). This approach aligns camera motion with human perception of directional movement. See Figure 18 for examples.

**Rotation.** The camera rotates along its own axis to the right or left (`pan`), up or down (`tilt`), or clockwise or counterclockwise (`roll`). Refer to Table 13 for details. Note: Pure camera rotation (without translation) does not produce a parallax effect. Take `pan-left` as an example: the entire

Table 5: **Evaluation of video-text retrieval models.** We compare all VLMs on text, video, and group score across all skills.

| Model | Motion & Steadiness | | | Scene Dynamics | | | Motion Speed | | | Motion Direction | | | Confusable Motion | | | Has Motion | | | Tracking Shot | | | Only Motion | | | Avg Overall | | |
|---|---|---|---|---|---|---|---|---|---|---|---|---|---|---|---|---|---|---|---|---|---|---|---|---|---|---|---|
| | T | V | G | T | V | G | T | V | G | T | V | G | T | V | G | T | V | G | T | V | G | T | V | G | T | V | G |
| Random Chance | 25.0 | 25.0 | 16.6 | 25.0 | 25.0 | 16.6 | 25.0 | 25.0 | 16.6 | 25.0 | 25.0 | 16.6 | 25.0 | 25.0 | 16.6 | 25.0 | 25.0 | 16.6 | 25.0 | 25.0 | 16.6 | 25.0 | 25.0 | 16.6 | 25.0 | 25.0 | 16.6 |
| *CLIPScore* | | | | | | | | | | | | | | | | | | | | | | | | | | | |
| UMT-B16 | 25.0 | 3.0 | 2.4 | 21.8 | 3.5 | 0.0 | 36.8 | 2.3 | 2.3 | 23.3 | 0.2 | 0.2 | 27.1 | 1.7 | 0.6 | 31.1 | 6.9 | 4.8 | 24.2 | 5.8 | 3.6 | 15.8 | 1.4 | 0.7 | 26.8 | 4.1 | 2.8 |
| UMT-L16 | 15.9 | 2.0 | 1.4 | 12.6 | 2.3 | 2.3 | 40.2 | 3.5 | 3.5 | 21.9 | 1.3 | 0.5 | 18.6 | 2.8 | 0.6 | 27.8 | 7.5 | 4.6 | 27.3 | 4.6 | 3.0 | 13.0 | 2.2 | 0.0 | 23.7 | 4.4 | 2.6 |
| LanguageBind | 17.2 | 6.8 | 3.7 | 18.4 | 8.1 | 5.8 | 33.3 | 13.8 | 10.3 | 22.6 | 5.8 | 4.0 | 26.6 | 5.1 | 2.8 | 25.3 | 11.1 | 7.6 | 28.5 | 17.0 | 10.3 | 18.0 | 6.5 | 1.4 | 24.0 | 9.7 | 6.2 |
| LanguageBindV1.5 | 18.9 | 5.4 | 2.4 | 20.7 | 10.3 | 5.8 | 32.2 | 10.3 | 9.2 | 21.7 | 3.8 | 2.5 | 22.0 | 7.9 | 5.7 | 25.7 | 10.0 | 6.6 | 30.6 | 12.4 | 8.5 | 17.3 | 6.5 | 3.6 | 24.2 | 8.3 | 5.4 |
| InternVideo2-S2 | 1.4 | 3.0 | 0.0 | 32.2 | 3.5 | 3.5 | 2.3 | 3.5 | 1.2 | 9.6 | 0.0 | 0.0 | 8.5 | 4.5 | 2.3 | 13.4 | 2.1 | 0.8 | 1.2 | 3.6 | 0.3 | 8.6 | 2.2 | 0.7 | 9.3 | 2.3 | 0.7 |
| *ITMScore* | | | | | | | | | | | | | | | | | | | | | | | | | | | |
| UMT-B16 | 0.7 | 4.7 | 0.0 | 2.3 | 8.1 | 1.2 | 16.1 | 5.8 | 2.3 | 11.6 | 3.6 | 1.6 | 22.0 | 5.7 | 1.7 | 18.9 | 13.8 | 5.8 | 23.3 | 12.7 | 8.2 | 4.3 | 4.3 | 2.2 | 14.7 | 9.1 | 3.9 |
| UMT-L16 | 13.5 | 8.8 | 4.1 | 26.4 | 8.1 | 6.9 | 29.9 | 10.3 | 3.5 | 12.3 | 2.0 | 0.7 | 13.0 | 5.1 | 1.1 | 24.8 | 16.9 | 8.0 | 20.9 | 13.6 | 7.0 | 7.9 | 3.6 | 0.7 | 19.1 | 10.7 | 5.0 |
| InternVideo2-S2 | 18.2 | 9.8 | 6.1 | 6.9 | 10.3 | 2.3 | 37.9 | 6.9 | 4.6 | 7.4 | 2.9 | 0.7 | 29.9 | 7.9 | 4.5 | 21.3 | 11.7 | 4.5 | 19.1 | 10.0 | 7.3 | 9.4 | 2.9 | 1.4 | 18.2 | 8.7 | 4.1 |
| *VQAScore* | | | | | | | | | | | | | | | | | | | | | | | | | | | |
| mPLUG-Owl3-7B | 18.2 | 39.9 | 15.9 | 54.0 | 79.3 | 52.9 | 48.3 | 41.4 | 28.7 | 23.9 | 17.0 | 9.2 | 13.0 | 20.9 | 6.8 | 31.8 | 48.6 | 27.4 | 22.7 | 44.2 | 18.2 | 7.2 | 18.0 | 3.6 | 26.2 | 38.4 | 19.6 |
| LLaVA-Video-7B | 11.5 | 39.2 | 10.1 | 51.7 | 74.7 | 50.6 | 31.0 | 51.7 | 25.3 | 15.7 | 14.5 | 6.0 | 15.8 | 15.8 | 5.1 | 8.9 | 54.9 | 8.4 | 39.4 | 53.3 | 33.6 | 18.0 | 12.2 | 6.5 | 17.8 | 40.9 | 13.3 |
| LLaVA-OneVision-7B | 20.3 | 46.6 | 18.2 | 47.1 | 77.0 | 47.1 | 50.6 | 46.0 | 39.1 | 17.7 | 14.1 | 6.0 | 8.5 | 18.1 | 3.4 | 23.9 | 49.5 | 20.9 | 39.4 | 52.7 | 32.4 | 10.8 | 13.0 | 5.8 | 24.3 | 39.7 | 18.9 |
| InternVideo2-Chat-8B | 14.9 | 16.9 | 5.1 | 33.3 | 71.3 | 33.3 | 37.9 | 28.7 | 10.3 | 20.8 | 9.8 | 5.2 | 18.6 | 18.1 | 9.6 | 28.2 | 18.7 | 9.9 | 11.5 | 16.4 | 3.6 | 2.9 | 5.8 | 1.4 | 21.4 | 18.0 | 8.0 |
| InternVideo2-Chat-26B | 17.6 | 22.6 | 9.8 | 23.0 | 48.3 | 20.7 | 44.8 | 29.9 | 24.1 | 42.5 | 17.2 | 13.9 | 20.3 | 10.2 | 5.1 | 47.3 | 29.9 | 22.3 | 27.6 | 19.1 | 11.5 | 7.2 | 3.6 | 0.7 | 35.1 | 23.1 | 15.4 |
| InternLMXComposer2.5-7B | 32.1 | 43.6 | 25.7 | 9.2 | 69.0 | 8.1 | 31.0 | 44.8 | 28.7 | 22.8 | 17.5 | 10.1 | 11.9 | 19.2 | 6.2 | 7.5 | 37.4 | 6.8 | 5.5 | 32.7 | 2.7 | 10.8 | 19.4 | 4.3 | 14.3 | 33.0 | 9.8 |
| InternVL2.5-8B | 14.5 | 52.0 | 12.8 | 51.7 | 70.1 | 50.6 | 36.8 | 43.7 | 29.9 | 14.0 | 17.9 | 4.7 | 17.8 | 29.6 | 14.8 | 19.4 | 62.4 | 18.5 | 29.7 | 53.3 | 18.1 | 2.4 | 9.8 | 2.4 | 22.0 | 43.9 | 17.5 |
| InternVL2.5-26B | 15.5 | 53.4 | 15.2 | 55.2 | 77.0 | 55.2 | 50.6 | 62.1 | 47.1 | 15.5 | 18.3 | 8.6 | 4.4 | 26.7 | 4.4 | 29.6 | 57.3 | 27.7 | 42.2 | 62.3 | 34.2 | 0.0 | 14.6 | 0.0 | 22.1 | 45.1 | 18.7 |
| InternVL3-8B | 17.2 | 54.7 | 16.6 | 52.9 | 74.7 | 49.4 | 59.8 | 43.7 | 37.9 | 20.6 | 16.1 | 8.7 | 11.9 | 29.9 | 6.8 | 38.5 | 59.4 | 35.8 | 46.4 | 52.4 | 31.8 | 16.6 | 24.5 | 8.6 | 31.9 | 46.0 | 25.0 |
| InternVL3-78B | 21.6 | 58.3 | 19.4 | 55.7 | 76.2 | 52.1 | 63.1 | 45.4 | 39.8 | 24.2 | 19.5 | 10.3 | 15.8 | 33.7 | 9.6 | 42.3 | 61.8 | 39.2 | 49.7 | 54.9 | 35.4 | 18.3 | 27.1 | 10.4 | 35.7 | 48.6 | 27.8 |
| Qwen2.5-VL-7B | 29.1 | 55.7 | 24.7 | 41.4 | 72.4 | 40.2 | 59.8 | 50.6 | 33.3 | 20.8 | 18.8 | 9.8 | 16.4 | 27.7 | 10.7 | 43.9 | 60.2 | 40.7 | 46.4 | 13.0 | 7.6 | 13.0 | 10.1 | 2.9 | 35.0 | 40.8 | 24.2 |
| Qwen2.5-VL-32B | 43.6 | 63.9 | 42.9 | 37.9 | 70.1 | 37.9 | 65.5 | 50.6 | 36.8 | 34.0 | 23.7 | 15.0 | 26.6 | 17.0 | 10.7 | 47.5 | 59.3 | 43.7 | 46.1 | 23.9 | 15.5 | 15.1 | 5.8 | 2.9 | 41.4 | 42.7 | 29.5 |
| Qwen2.5-VL-72B | 46.5 | 67.1 | 45.3 | 39.2 | 70.8 | 38.5 | 67.8 | 51.3 | 38.6 | 37.3 | 26.4 | 16.8 | 30.2 | 20.8 | 12.4 | 49.3 | 60.7 | 45.2 | 47.5 | 27.8 | 17.6 | 16.2 | 8.3 | 3.8 | 43.7 | 44.5 | 32.1 |
| GPT-4o | 27.4 | 43.2 | 22.6 | 2.3 | 73.6 | 2.3 | 52.9 | 46.0 | 28.7 | 40.7 | 34.7 | 26.0 | 33.3 | 29.9 | 17.5 | 43.2 | 54.7 | 36.2 | 45.5 | 26.7 | 16.4 | 24.5 | 16.0 | 10.1 | 38.3 | 42.4 | 25.8 |
| **Qwen2.5-VL-7B (SFT)** | 45.5 | 59.2 | 45.5 | 53.0 | 65.8 | 53.0 | 53.8 | 63.4 | 53.8 | 83.8 | 98.4 | 74.6 | 61.7 | 57.5 | 36.7 | 83.9 | 125.8 | 83.2 | 53.3 | 66.3 | 52.7 | 43.6 | 66.5 | 43.6 | 44.6 | 59.1 | 42.7 |
| **Qwen2.5-VL-32B (SFT)** | 50.3 | 60.9 | 50.1 | 52.6 | 65.8 | 52.3 | 55.4 | 62.8 | 53.8 | 91.4 | 97.7 | 78.2 | 70.3 | 62.3 | 40.9 | 82.4 | 119.9 | 82.2 | 54.8 | 67.2 | 54.1 | 44.6 | 67.1 | 44.6 | 45.8 | 61.2 | 43.9 |
| **Qwen2.5-VL-72B (SFT)** | 52.0 | 61.9 | 51.6 | 53.1 | 65.1 | 52.9 | 55.7 | 62.8 | 54.4 | 93.6 | 99.2 | 80.4 | 71.8 | 61.5 | 39.3 | 84.7 | 122.1 | 84.4 | 54.4 | 68.5 | 54.3 | 45.3 | 68.1 | 45.1 | 46.3 | 62.2 | 44.4 |

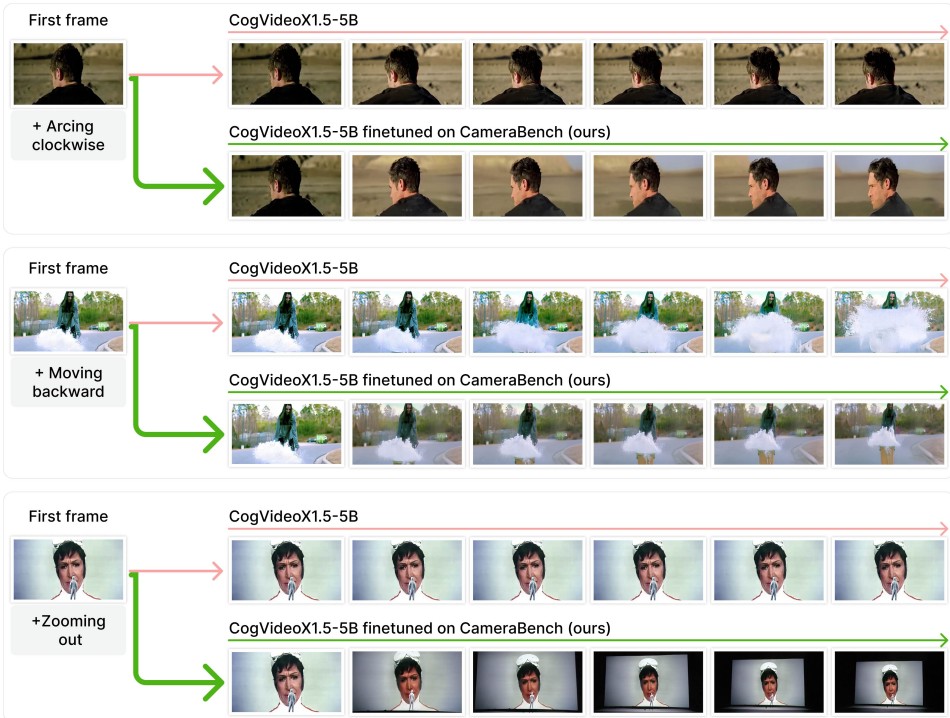

Figure 17: **Fine-tuning CogVideoX-1.5 on CameraBench improves motion control.** We show three random test examples comparing the original CogVideoX and our LoRA fine-tuned model. Fine-tuning on CameraBench's motion-rich captions improves the model's ability to follow motion instructions like dolly, zoom, and arc.

scene appears to rotate leftward, but the relative positions of objects remain unchanged. In contrast, for `truck-left`, closer objects move faster due to camera translation.

**Intrinsic change.** The camera adjusts its focal length to zoom in or out (`zoom`). Refer to Table 14 for details. Pure camera zooming (without translation) does not create a parallax effect; it magnifies the scene while preserving object positions, making the scene appear to scale around the optical center.

Table 6: **SFT hyperparameters** for Qwen-2.5-VL-7B.

| Hyperparameter | Value |
|---|---|
| finetuning_type | full |
| per_device_train_batch_size | 4 |
| gradient_accumulation_steps | 2 |
| learning_rate | 2.0e-5 |
| num_train_epochs | 6.0 |
| lr_scheduler_type | cosine |
| warmup_ratio | 0.05 |
| freeze_vision_tower | true |
| freeze_multi_modal_projector | true |
| video_fps | 8.0 |
| video_max_pixels | 16384 |
| image_max_pixels | 262144 |
| deepspeed | ds_z3_config.json |
| template | qwen2_vl |
| bf16 | true |
| flash_attn | fa2 |

Table 7: **SFT hyperparameters** for Qwen-2.5-VL-32B.

| Hyperparameter | Value |
|---|---|
| finetuning_type | full |
| per_device_train_batch_size | 1 |
| gradient_accumulation_steps | 2 |
| learning_rate | 1.0e-5 |
| num_train_epochs | 6.0 |
| lr_scheduler_type | cosine |
| warmup_ratio | 0.05 |
| freeze_vision_tower | true |
| freeze_multi_modal_projector | true |
| video_fps | 8.0 |
| video_max_pixels | 16384 |
| image_max_pixels | 262144 |
| deepspeed | ds_z3_config.json |
| template | qwen2_vl |
| bf16 | true |
| flash_attn | fa2 |

In contrast, camera translation introduces parallax, causing closer objects to change size within the frame more quickly.

**Object-centric movements.** The camera orbits around a subject (or the frame center) in a circular path (`arc`), or tracks a moving subject from behind (`tail-tracking`), the front (`lead-tracking`), the side (`side-tracking`), from an aerial view (`aerial-tracking`), or using other motions (`tilt-/pan-/arc-tracking`). We also consider whether the camera moves or zooms to make the subject appear `larger` or `smaller` within the frame. Refer to Table 15 for details.

**Others.** We include the speed of camera movement (`slow/regular/fast`), motion effects (`dolly-zoom/motion-blur`), and scene movement (`static/mostly-static/dynamic`). Refer to Table 17 for details.

# G   Skills and Tasks in CameraBench

**Skills, tasks, and their textual definitions.** We detail all 9 top-level skills and their 81 sub-tasks in Table 18. Additionally, we report the textual definitions used to construct the prompts for VLMs.

Table 8: **SFT hyperparameters** for Qwen-2.5-VL-72B.

| Hyperparameter | Value |
|---|---|
| finetuning_type | full |
| per_device_train_batch_size | 1 |
| gradient_accumulation_steps | 2 |
| learning_rate | 1.0e-5 |
| num_train_epochs | 6.0 |
| lr_scheduler_type | cosine |
| warmup_ratio | 0.05 |
| freeze_vision_tower | true |
| freeze_multi_modal_projector | true |
| video_fps | 8.0 |
| video_max_pixels | 16384 |
| image_max_pixels | 262144 |
| deepspeed | ds_z3_config.json |
| template | qwen2_vl |
| bf16 | true |
| flash_attn | fa2 |

Table 9: **FPS/SFT ablations**. We report Average Precision (AP) for binary classification of camera-centric motion primitives. Our results show that higher FPS generally improves performance. Additionally, full fine-tuning of Qwen-2.5-7B outperforms LoRA-based fine-tuning.

| Model/FPS | Translation (Dolly/Pedestal/Truck) | | | | | | Zooming | | Rotation (Pan/Tilt/Roll) | | | | | | Static | Avg |
|---|---|---|---|---|---|---|---|---|---|---|---|---|---|---|---|---|
| | In | Out | Up | Down | Right | Left | In | Out | Right | Left | Up | Down | CW | CCW | | |
| *MegaSAM* | | | | | | | | | | | | | | | | |
| 2 FPS | 65.9 | 43.3 | 19.4 | 21.3 | 36.6 | 35.8 | 11.1 | 10.2 | 62.9 | 75.8 | 68.2 | 59.5 | 73.1 | 85.9 | 19.6 | 45.9 |
| 4 FPS | 72.7 | 42.6 | 23.0 | 31.8 | 44.6 | 39.9 | 11.1 | 10.2 | 72.6 | 78.8 | 79.0 | 60.9 | 72.5 | 70.4 | 24.4 | 49.0 |
| 8 FPS | **75.0** | 43.4 | **27.6** | **42.8** | **46.2** | 39.9 | 11.1 | 10.2 | 77.9 | **82.4** | **75.6** | 57.6 | 67.3 | **76.8** | 19.7 | 50.2 |
| 30 FPS | 73.8 | **43.9** | 24.2 | 29.1 | 45.3 | **44.2** | 11.1 | 10.2 | **79.5** | 82.2 | 73.8 | **65.3** | 71.5 | 75.8 | **22.0** | 50.1 |
| *Qwen-2.5-LoRA-SFT* | | | | | | | | | | | | | | | | |
| 2 FPS | 76.9 | 37.6 | 12.3 | 26.6 | 58.6 | 36.9 | 46.3 | 62.1 | 72.7 | 82.2 | 68.2 | 57.0 | 32.6 | 37.4 | 63.0 | 51.3 |
| 4 FPS | 78.6 | 40.4 | 15.1 | 29.8 | 61.0 | 39.6 | 49.1 | 65.2 | 75.6 | 84.3 | 69.9 | 59.7 | 35.3 | 40.2 | 66.2 | 54.2 |
| 8 FPS | 81.3 | 43.1 | 16.9 | 32.2 | 62.5 | 42.3 | 50.8 | 68.3 | 77.5 | 86.4 | 73.2 | 60.6 | 37.5 | 43.7 | 68.1 | **56.7** |
| *Qwen-2.5-Full-SFT* | | | | | | | | | | | | | | | | |
| 2 FPS | 78.2 | 42.7 | 22.2 | 41.9 | 56.3 | 48.5 | 45.2 | 63.5 | 71.9 | 82.6 | 65.4 | 52.9 | 33.6 | 41.3 | 61.2 | 56.8 |
| 4 FPS | 80.3 | 46.0 | 24.8 | 47.6 | 61.3 | 52.0 | 48.8 | 68.5 | 74.7 | 83.6 | 67.7 | 55.9 | 37.7 | 45.7 | 63.3 | 58.4 |
| 8 FPS | **83.2** | **48.6** | **27.2** | **48.8** | **62.6** | **54.3** | **51.3** | **70.7** | **77.6** | **86.9** | **70.4** | **58.0** | **38.5** | **46.3** | **65.2** | **59.3** |

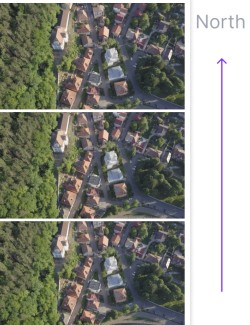

Figure 18: We define moving forward (`dolly-in`) for a bird's-eye view camera in a ground-centric reference frame as movement toward the north (the top of the frame) to maintain label consistency.

Table 10: **Motion type** definitions and guidelines.

| Motion Type | Options | Definition |
|---|---|---|
| | no-motion | The camera remains stationary with no intentional movement. Note: Unintentional shaking belongs to "no motion". |
| | minor-motion | The camera moves slightly and intentionally, such as a gentle pan or zoom. The motion is noticeable but remains subtle and not significant. |
| | simple-motion | The camera moves significantly in a straightforward manner, such as a steady pan, tilt, arc, or simple tracking shot. Note: Select this even if the video combines two or more motions, as long as they occur simultaneously at roughly the same speed. |
| Motion Type | complex-motion | The camera exhibits complex movements that are difficult to classify. This includes: (1) Conflicting Motion: Opposing movements occur, such as panning left then right, often seen in drone maneuvers, video game shots, or fast-paced action scenes. (2) Sequential Motion: Two or more movements happen one after another rather than simultaneously (e.g., moving forward, then shifting position after stopping). (3) Simultaneous Motions at Different Speeds: Multiple simultaneous movements occur at significantly different speeds. (4) Unclear Motion / Missing Background Information: If the motion is difficult to analyze due to motion blur or lack of background cues. |

Table 11: **Steadiness** definitions and guidelines.

| Steadiness | Options | Definition |
|---|---|---|
| | static | The camera remains completely stationary with no movement or vibration. |
| | no-shaking | The camera moves smoothly with no detectable shake, typically using high-end stabilizers. Select only if (1) the camera is moving and (2) no unintended motion is present. |
| Steadiness | minimal-shaking | The camera exhibits slight shaking, whether stationary or moving, maintaining a mostly stable shot. Select even if stationary with slight shake. Note: Select even if stationary with slight shake. |
| | unsteady | The camera shows moderate shaking, whether stationary or in motion, introducing noticeable but controlled instability. Note: Select even if stationary with noticeable shake. |
| | very unsteady | The camera shakes consistently, typical of unstabilized handheld or action footage. Note: Select only if shaking is consistent throughout the video. |

Table 12: **Camera translation** definitions and guidelines.

| Translation | Options | Definition |
|---|---|---|
| **Dolly** | dolly-in/ dolly-out | The camera moves forward or backward relative to the ground plane and the initial frame. |
| | no-dolly | The camera does not move forward/backward during the shot. |
| **Pedestal** | pedestal-up/ pedestal-down | Select this when the camera moves upward or downward clearly and consistently relative to the ground or the orientation of the initial frame. |
| | no-pedestal | Select this label when the camera does not move leftward/rightward during the shot. |
| **Truck** | truck-left/ truck-right | The camera physically moves to the left or right, changing its position relative to the initial frame. |
| | no-truck | The camera does not move to the left or right during the shot. |

Table 13: **Camera rotation** definitions and guidelines.

| Rotation | Options | Definition |
|---|---|---|
| **Pan** | `pan-left/` `pan-right` | The camera rotates its angle by pivoting left or right with respect to the initial frame. |
| | `no-pan` | The camera does not pan left or right. |
| **Tilt** | `tilt-up/` `tilt-down` | The camera rotates its angle up or down vertically with respect to the initial frame. |
| | `no-tilt` | The camera does not tilt up or down. |
| **Roll** | `roll-CW/` `roll-CCW` | The camera performs a clear and consistent clockwise (CW) or counterclockwise (CCW) roll by rotating around its own optical center. |
| | `no-roll` | The camera does not roll clockwise/counterclockwise. |

Table 14: **Camera intrinsic change** definitions and guidelines.

| Zooming | Options | Definition |
|---|---|---|
| **Zoom** | `zoom-in/` `zoom-out` | The camera adjusts its focal length to zoom in or out, changing the frame size. Note: This differs from physical camera movement. |
| | `no-zoom` | The camera does not adjust its focal length during the video. |

Table 15: **Object-centric movement** definitions and guidelines.

| Object-centric Motion | Options | Definition |
| --- | --- | --- |
| Arc | `arc-CW/`
`arc-CCW` | The camera moves in a circular or semi-circular motion around the subject (or the frame center) in a clockwise or counterclockwise direction. |
| | `no-arc` | The camera does not move in a circular or semi-circular motion during the video. |
| Arc-Tracking | `arc-tracking` | The camera moves in a circular or semi-circular path around the moving subject, often referred to as an orbit or circular tracking shot. |
| | `no-arc-tracking` | The camera does not track or does not move in a circular or semi-circular path around the moving subject. |
| Lead-Tracking | `lead-tracking` | The camera moves ahead of the moving subject, capturing their face or front as they follow the camera's path. This is also referred to as a leading shot. |
| | `no-lead-tracking` | The camera does not track or does not move ahead of the moving subject. |
| Tail-Tracking | `tail-tracking` | The camera follows directly behind the moving subject, keeping their back in view as they move forward. This is also known as a follow shot or chase shot. |
| | `no-tail-tracking` | The camera does not track or does not move behind the moving subject. |
| Side-Tracking | `side-tracking` | The camera moves parallel to the moving subject, following them from the side as they move through the scene. This is often referred to as a trucking shot in film terminology. |
| | `no-side-tracking` | The camera does not track or does not move parallel to the moving subject. |
| Aerial-Tracking | `aerial-tracking` | The camera tracks the moving subject from a high vantage point, often using a drone or crane to follow their movement. |
| | `no-aerial-tracking` | The camera either does not track the moving subject or is not positioned at a high vantage point. |
| Pan-Tracking | `pan-tracking` | The camera remains in a fixed position but pivots horizontally to follow the subject as they move. |
| | `no-pan-tracking` | The camera does not track the subject or does not pivot horizontally to follow their movement. |
| Tilt-Tracking | `tilt-tracking` | The camera tilts up or down to follow the vertical movement of the subject. |
| | `no-tilt-tracking` | The camera does not track the subject or does not pivot vertically to follow their movement. |
| Subject Size Change | `subject-larger` | The camera moves or zooms in towards the tracked subject, making them appear larger in the frame. |
| | `subject-smaller` | The camera moves or zooms away from the tracked subject, making them appear smaller in the frame. |
| | `no-subject-change` | The camera neither moves towards nor away from the subject. |

Table 16: **Camera movement speed** definitions and guidelines.

| Motion Speed | Options | Definition |
| --- | --- | --- |
| Moving Speed | `slow` | The camera moves at a noticably slow pace. |
| | `regular` | The camera moves at a regular pace. If the speed does not stand out as particularly slow or fast, it is considered regular. |
| | `fast` | The camera moves quickly, such as in a crash zoom or whip pan. |

Table 17: **Others** definitions and guidelines.

| Others | Options | Definition |
|---|---|---|
| **Camera Movement Speed** | slow | The camera moves at a noticeably slow pace. |
| | regular | The camera moves at a regular pace. If the speed does not stand out as particularly slow or fast, it is considered regular. |
| | fast | The camera moves quickly, such as in a crash zoom or whip pan. |
| **Cinematic Motion Effects** | frame-freezing | A visual effect where scene motion is paused or frozen mid-action, creating a still frame within a moving sequence. |
| | dolly-zoom | A camera effect where the background appears to compress or stretch while the subject stays the same size, often used to create a sense of unease. |
| | motion-blur | A visual effect where moving objects blur due to slow shutter speed or camera movement, often used to emphasize speed and fluid motion in action scenes. |
| **Scene Dynamics** | static | The entire scene, including all subjects and background, remains completely motionless throughout the video. |
| | mostly-static | The scene is largely still, with only minor elements or small parts exhibiting movement. |
| | dynamic | A significant portion of the frame is occupied by dynamic movement of subjects or scene elements (excluding camera motion) that visibly alters the scene. |

Table 18: **All tasks for each top-level skill in CameraBench.** We list all 81 tasks of 9 skills in CameraBench.

| Skill | Description | Tasks |
|---|---|---|
| **Motion & Steadiness** | Evaluates how steady the camera is and whether it moves in a controlled manner, including shake detection and fixed vs. moving camera states. | Clear Moving Camera, Fixed Camera Shake, Stable vs. Shaky Camera, Fixed vs. Moving Camera. (4 Tasks in Table 19) |
| **Scene Dynamics** | Determines whether a scene is static or dynamic, and detects frame freeze effects. | Static vs. Dynamic Scene, Frame Freeze Effect. (2 Tasks in Table 20) |
| **Motion Speed** | Evaluates the speed of camera movements, distinguishing between slow-moving and fast-moving shots, and detects motion blur. | Slow vs. Fast Movement, Motion Blur Effect. (2 Tasks in Table 16) |
| **Motion Direction** | Classifies the direction of camera motion, including forward/backward, upward/downward, leftward/rightward, panning, tilting, rolling, and complex movement types like crane and arc shots. | Dolly In vs. Out (Ground), Pedestal Up vs. Down (Ground), Truck Left vs. Right, Pan Left vs. Right, Tilt Up vs. Down, Roll CW vs. CCW, Side Tracking Left vs. Right, Lead vs. Tail Tracking, Arc CCW vs. CW, Crane Up vs. Down, Dolly Zoom In vs. Out, Zoom In vs. Out. (12 Tasks in Table 22) |
| **Confusable Motion** | Distinguishes between commonly confused motion types, such as zooming versus physical movement, translation versus rotation, and differentiating the reference frame in which the motion happens. | Zoom In vs. Dolly In, Zoom Out vs. Dolly Out, Only Zoom In vs. Only Dolly In, Only Zoom Out vs. Only Dolly Out, Pan Right vs. Truck Right, Pan Left vs. Truck Left, Only Pan Right vs. Only Truck Right, Only Pan Left vs. Only Truck Left, Tilt Up vs. Pedestal Up, Tilt Down vs. Pedestal Down, Only Tilt Up vs. Only Pedestal Up, Only Tilt Down vs. Only Pedestal Down, Dolly In Camera vs. Ground, Dolly Out Camera vs Ground, Pedestal Up Camera vs. Ground, Pedestal Down Camera vs. Ground. (16 Tasks in Table 23) |
| **Has Motion** | Determines whether the camera exhibits motion, including intrinsic changes (zoom) and physical movement (translation, rotation, or arc motion). | Zoom In, Zoom Out, Dolly In, Dolly Out, Pedestal Up, Pedestal Down, Truck Right, Truck Left, Pan Right, Pan Left, Tilt Up, Tilt Down, Roll CW, Roll CCW, Arc CW, Arc CCW, Crane Up, Crane Down. (18 Tasks in Table 24) |
| **Tracking Shot** | Identifies whether the camera is tracking a subject, specifies different types of tracking shots. | General Tracking, Aerial Tracking, Arc Tracking, Front-Side Tracking, Rear-Side Tracking, Lead Tracking, Tail Tracking, Tilt Tracking, Pan Tracking, Side Tracking, Tracking Subject Larger, Tracking Subject Smaller. (12 Tasks in Table 25) |
| **Only Motion** | Identifies cases where the camera performs a single motion type without any other movement. | Only Zoom In, Only Zoom Out, Only Dolly In, Only Dolly Out, Only Pedestal Up, Only Pedestal Down, Only Truck Right, Only Truck Left, Only Pan Right, Only Pan Left, Only Tilt Up, Only Tilt Down, Only Roll CW, Only Roll CCW. (14 Tasks in Table 26) |
| **Complex Description** | Determines whether a given motion description correctly describes the camera movement in a video. | Complex Description. (1 Task) |

Table 19: **Motion & Steadiness Tasks**

| Tasks | Questions | Descriptions |
|---|---|---|
| **Clear Moving Camera** | **Positive:** Does the camera have noticeable motion beyond minor shake or wobble? | **Positive:** A video where the camera has noticeable motion beyond minor shake or wobble. |
| | **Negative:** Is the camera free from noticeable motion beyond minor shake or wobble? | **Negative:** A video where the camera is free from noticeable motion beyond minor shake or wobble. |
| **Fixed Camera Shake** | **Positive:** Is the camera completely still without any motion or shaking? | **Positive:** A video where the camera remains completely still with no motion or shaking. |
| | **Negative:** Is the camera stationary with minor vibrations or shaking? | **Negative:** A video where the camera is mostly stationary but has minor vibrations or shaking. |
| **Stable vs. Shaky Camera** | **Positive:** Is the camera movement exceptionally smooth and highly stable? | **Positive:** A video where the camera movement is exceptionally smooth and highly stable. |
| | **Negative:** Does the camera show noticeable vibrations, shaking, or wobbling? | **Negative:** A video where the camera shows noticeable vibrations, shaking, or wobbling. |
| **Fixed vs. Moving Camera** | **Positive:** Is the camera completely still without any visible movement? | **Positive:** The camera is completely still without any visible movement. |
| | **Negative:** Is the camera not completely still and shows visible movement? | **Negative:** The camera is not completely still and shows visible movement. |

Table 20: **Scene Dynamics Tasks**

| Tasks | Questions | Descriptions |
|---|---|---|
| **Static vs. Dynamic Scene** | **Positive:** Is the scene in the video completely static? | **Positive:** A video where the scene is completely static. |
| | **Negative:** Is the scene in the video dynamic? | **Negative:** A video where the scene is dynamic and features movement. |
| **Frame Freeze Effect** | **Positive:** Does the video contain a frame freeze effect at any point? | **Positive:** A video that contains a frame freeze effect at some point. |
| | **Negative:** Is the video free from any frame freeze effect? | **Negative:** A video that is free from any frame freeze effect. |

Table 21: **Camera Motion Speed Tasks**

| Tasks | Questions | Descriptions |
|---|---|---|
| **Slow vs. Fast Movement** | **Positive:** Does the camera have noticeable motion but at a slow motion speed? | **Positive:** A video where the camera has noticeable motion at a slow speed. |
| | **Negative:** Does the camera have noticeable motion but at a fast motion speed? | **Negative:** A video where the camera has noticeable motion at a fast speed. |
| **Motion Blur Effect** | **Positive:** Does the video contain noticeable motion blur? | **Positive:** The video exhibits a motion blur effect. |
| | **Negative:** Is the video free from any noticeable motion blur? | **Negative:** The video is free from any noticeable motion blur. |

Table 22: **Camera Motion Direction Tasks**

| Tasks | Questions | Descriptions |
|---|---|---|
| **Dolly In vs. Out (Ground)** | **Positive:** Is the camera moving forward in the scene? | **Positive:** A shot where the camera is moving forward within the scene. |
| | **Negative:** Is the camera moving backward in the scene? | **Negative:** A shot where the camera is moving backward within the scene. |
| **Pedestal Up vs. Down (Ground)** | **Positive:** Does the camera move upward relative to the ground? | **Positive:** The camera is moving upward relative to the ground. |
| | **Negative:** Does the camera move downward relative to the ground? | **Negative:** The camera is moving downward relative to the ground. |
| **Truck Left vs. Right** | **Positive:** Does the camera move leftward in the scene? | **Positive:** The camera moves leftward. |
| | **Negative:** Does the camera move rightward in the scene? | **Negative:** The camera moves rightward. |
| **Pan Left vs. Right** | **Positive:** Does the camera pan to the left? | **Positive:** The camera pans to the left. |
| | **Negative:** Does the camera pan to the right? | **Negative:** The camera pans to the right. |
| **Tilt Up vs. Down** | **Positive:** Does the camera tilt upward? | **Positive:** The camera tilts upward. |
| | **Negative:** Does the camera tilt downward? | **Negative:** The camera tilts downward. |
| **Roll CW vs. CCW** | **Positive:** Does the camera roll clockwise? | **Positive:** The camera rolls clockwise. |
| | **Negative:** Does the camera roll counterclockwise? | **Negative:** The camera rolls counterclockwise. |
| **Side Tracking Left vs. Right** | **Positive:** Is it a side-tracking shot where the camera moves left to follow the subject? | **Positive:** A side-tracking shot where the camera moves left to follow the subject. |
| | **Negative:** Is it a side-tracking shot where the camera moves right to follow the subject? | **Negative:** A side-tracking shot where the camera moves right to follow the subject. |
| **Lead vs. Tail Tracking** | **Positive:** Is it a tracking shot with the camera moving ahead of the subject? | **Positive:** A tracking shot where the camera moves ahead of the subject. |
| | **Negative:** Is it a tracking shot with the camera following behind the subject? | **Negative:** A tracking shot where the camera follows behind the subject. |
| **Arc CCW vs. CW** | **Positive:** Does the camera move in a counterclockwise arc? | **Positive:** The camera arcs counterclockwise. |
| | **Negative:** Does the camera move in a clockwise arc? | **Negative:** The camera arcs clockwise. |
| **Crane Up vs. Down** | **Positive:** Is the camera craning upward in an arc? | **Positive:** The camera cranes upward in an arc. |
| | **Negative:** Does the camera move downward in a crane shot? | **Negative:** The camera cranes downward in an arc. |
| **Dolly Zoom In vs. Out** | **Positive:** Does the shot feature a dolly zoom effect with the camera moving backward and zooming in? | **Positive:** The camera performs a dolly zoom effect with backward movement and zoom-in. |
| | **Negative:** Does the shot feature a dolly zoom effect with the camera moving forward and zooming out? | **Negative:** The camera performs a dolly zoom effect with forward movement and zoom-out. |
| **Zoom In vs. Out** | **Positive:** Does the camera zoom in? | **Positive:** The camera zooms in. |
| | **Negative:** Does the camera zoom out? | **Negative:** The camera zooms out. |

Table 23: **Confusable Motion Tasks**

| Tasks | Questions | Descriptions |
|---|---|---|
| **Zoom In vs. Dolly In** | **Positive:** Does the camera zoom in without physically moving forward? | **Positive:** A video where the camera zooms in without physically moving forward. |
| | **Negative:** Does the camera physically move forward without zooming in? | **Negative:** A video where the camera physically moves forward without zooming in. |
| **Zoom Out vs. Dolly Out** | **Positive:** Does the camera zoom out without physically moving backward? | **Positive:** A video where the camera zooms out without physically moving backward. |
| | **Negative:** Does the camera physically move backward without zooming out? | **Negative:** A video where the camera physically moves backward without zooming out. |
| **Only Zoom In vs. Only Dolly In** | **Positive:** Does the camera only zoom in without any other camera movement? | **Positive:** A video where the camera only zooms in with no other movement. |
| | **Negative:** Does the camera only move forward without any other camera movement? | **Negative:** A video where the camera only moves forward with no other movement. |
| **Only Zoom Out vs. Only Dolly Out** | **Positive:** Does the camera only zoom out without any other camera movement? | **Positive:** A video where the camera only zooms out with no other movement. |
| | **Negative:** Does the camera only move backward without any other camera movement? | **Negative:** A video where the camera only moves backward with no other movement. |
| **Pan Right vs. Truck Right** | **Positive:** Does the camera pan right without moving laterally to the right? | **Positive:** The camera pans right without moving laterally to the right. |
| | **Negative:** Does the camera move laterally to the right without panning right? | **Negative:** The camera moves laterally to the right without panning right. |
| **Pan Left vs. Truck Left** | **Positive:** Does the camera pan left without moving laterally to the left? | **Positive:** The camera pans left without moving laterally to the left. |
| | **Negative:** Does the camera move laterally to the left without panning left? | **Negative:** The camera moves laterally to the left without panning left. |
| **Only Pan Right vs. Only Truck Right** | **Positive:** Does the camera only pan right with no other movement? | **Positive:** A video where the camera only pans right with no other movement. |
| | **Negative:** Does the camera only move laterally to the right with no other movement? | **Negative:** A video where the camera only moves laterally to the right with no other movement. |
| **Only Pan Left vs. Only Truck Left** | **Positive:** Does the camera only pan left with no other movement? | **Positive:** A video where the camera only pans left with no other movement. |
| | **Negative:** Does the camera only move laterally to the left with no other movement? | **Negative:** A video where the camera only moves laterally to the left with no other movement. |
| **Tilt Up vs. Pedestal Up** | **Positive:** Does the camera tilt up without moving physically upward? | **Positive:** The camera tilts up without physically moving upward. |
| | **Negative:** Does the camera move physically upward without tilting up? | **Negative:** The camera moves physically upward without tilting up. |
| **Tilt Down vs. Pedestal Down** | **Positive:** Does the camera tilt down without moving physically downward? | **Positive:** The camera tilts down without physically moving downward. |
| | **Negative:** Does the camera move physically downward without tilting down? | **Negative:** The camera moves physically downward without tilting down. |
| **Only Tilt Up vs. Only Pedestal Up** | **Positive:** Does the camera only tilt up with no other movement? | **Positive:** A video where the camera only tilts up with no other movement. |
| | **Negative:** Does the camera only move physically upward with no other movement? | **Negative:** A video where the camera only moves physically upward with no other movement. |
| **Only Tilt Down vs. Only Pedestal Down** | **Positive:** Does the camera only tilt down with no other movement? | **Positive:** A video where the camera only tilts down with no other movement. |
| | **Negative:** Does the camera only move physically downward with no other movement? | **Negative:** A video where the camera only moves physically downward with no other movement. |
| **Dolly In Camera vs. Ground** | **Positive:** Does the camera move forward only relative to its initial viewing direction but not relative to the ground? | **Positive:** The camera moves forward only relative to its initial viewing direction but not relative to the ground. |
| | **Negative:** Does the camera move forward relative to both the ground and its initial viewing direction? | **Negative:** The camera moves forward relative to both the ground and its initial viewing direction. |
| **Dolly Out Camera vs Ground** | **Positive:** Does the camera move backward only relative to its initial viewing direction but not relative to the ground? | **Positive:** The camera moves backward only relative to its initial viewing direction but not relative to the ground. |
| | **Negative:** Does the camera move backward relative to both the ground and its initial viewing direction? | **Negative:** The camera moves backward relative to both the ground and its initial viewing direction. |
| **Pedestal Up Camera vs. Ground** | **Positive:** Does the camera move upward only relative to its initial viewing direction but not relative to the ground? | **Positive:** The camera moves upward only relative to its initial viewing direction but not relative to the ground. |
| | **Negative:** Does the camera move upward relative to both the ground and its initial viewing direction? | **Negative:** The camera moves upward relative to both the ground and its initial viewing direction. |
| **Pedestal Down Camera vs. Ground** | **Positive:** Does the camera move downward only relative to its initial viewing direction but not relative to the ground? | **Positive:** The camera moves downward only relative to its initial viewing direction but not relative to the ground. |
| | **Negative:** Does the camera move downward relative to both the ground and its initial viewing direction? | **Negative:** The camera moves downward relative to both the ground and its initial viewing direction. |

Table 24: **Has Motion Tasks**

| Tasks | Questions | Descriptions |
|---|---|---|
| **Zoom In** | **Positive:** Does the camera zoom in? | **Positive:** The camera zooms in. |
| | **Negative:** Is the camera free from any zoom in effects? | **Negative:** The camera is free from any zoom in effects. |
| **Zoom Out** | **Positive:** Does the camera zoom out? | **Positive:** The camera zooms out. |
| | **Negative:** Is the camera free from any zoom out effects? | **Negative:** The camera is free from any zoom out effects. |
| **Dolly In** | **Positive:** Is the camera moving forward in the scene? | **Positive:** The camera is moving forward within the scene. |
| | **Negative:** Is the camera free from any forward motion? | **Negative:** The camera is free from any forward motion. |
| **Dolly Out** | **Positive:** Is the camera moving backward in the scene? | **Positive:** The camera is moving backward within the scene. |
| | **Negative:** Is the camera free from any backward motion? | **Negative:** The camera is free from any backward motion. |
| **Truck Left** | **Positive:** Does the camera move laterally to the left? | **Positive:** The camera moves laterally to the left. |
| | **Negative:** Is the camera free from any leftward lateral movement? | **Negative:** The camera is free from any leftward lateral movement. |
| **Truck Right** | **Positive:** Does the camera move laterally to the right? | **Positive:** The camera moves laterally to the right. |
| | **Negative:** Is the camera free from any rightward lateral movement? | **Negative:** The camera is free from any rightward lateral movement. |
| **Pedestal Up** | **Positive:** Does the camera move upward relative to the ground? | **Positive:** The camera moves upward relative to the ground. |
| | **Negative:** Is the camera free from any upward pedestal motion? | **Negative:** The camera is free from any upward pedestal motion. |
| **Pedestal Down** | **Positive:** Does the camera move downward relative to the ground? | **Positive:** The camera moves downward relative to the ground. |
| | **Negative:** Is the camera free from any downward pedestal motion? | **Negative:** The camera is free from any downward pedestal motion. |
| **Pan Left** | **Positive:** Does the camera pan to the left? | **Positive:** The camera pans to the left. |
| | **Negative:** Is the camera free from any leftward panning motion? | **Negative:** The camera is free from any leftward panning motion. |
| **Pan Right** | **Positive:** Does the camera pan to the right? | **Positive:** The camera pans to the right. |
| | **Negative:** Is the camera free from any rightward panning motion? | **Negative:** The camera is free from any rightward panning motion. |
| **Tilt Up** | **Positive:** Does the camera tilt upward? | **Positive:** The camera tilts upward. |
| | **Negative:** Is the camera free from any upward tilting motion? | **Negative:** The camera is free from any upward tilting motion. |
| **Tilt Down** | **Positive:** Does the camera tilt downward? | **Positive:** The camera tilts downward. |
| | **Negative:** Is the camera free from any downward tilting motion? | **Negative:** The camera is free from any downward tilting motion. |
| **Roll CW** | **Positive:** Does the camera roll clockwise? | **Positive:** The camera rolls clockwise. |
| | **Negative:** Is the camera free from any clockwise rolling motion? | **Negative:** The camera is free from any clockwise rolling motion. |
| **Roll CCW** | **Positive:** Does the camera roll counterclockwise? | **Positive:** The camera rolls counterclockwise. |
| | **Negative:** Is the camera free from any counterclockwise rolling motion? | **Negative:** The camera is free from any counterclockwise rolling motion. |
| **Arc CW** | **Positive:** Does the camera move in a clockwise arc? | **Positive:** The camera moves in a clockwise arc. |
| | **Negative:** Is the camera free from any clockwise arc movement? | **Negative:** The camera is free from any clockwise arc movement. |
| **Arc CCW** | **Positive:** Does the camera move in a counterclockwise arc? | **Positive:** The camera moves in a counterclockwise arc. |
| | **Negative:** Is the camera free from any counterclockwise arc movement? | **Negative:** The camera is free from any counterclockwise arc movement. |

Table 25: **Tracking Shot Tasks**

| Tasks | Questions | Descriptions |
|---|---|---|
| **General Tracking** | **Positive:** Does the camera track the subject as they move? | **Positive:** The camera tracks the subject as they move. |
| | **Negative:** Is the video not a tracking shot? | **Negative:** The video is not a tracking shot. |
| **Aerial Tracking** | **Positive:** Does the camera track the subject from an aerial perspective? | **Positive:** The camera tracks the subject from an aerial perspective. |
| | **Negative:** Is the video not a tracking shot from an aerial perspective? | **Negative:** The camera is not tracking the subject from an aerial perspective. |
| **Arc Tracking** | **Positive:** Does the camera follow the subject while moving in an arc? | **Positive:** A tracking shot where the camera follows the subject while moving in an arc. |
| | **Negative:** Is the video not a tracking shot with arc movement? | **Negative:** The camera is not tracking the subject with arc movement. |
| **Front-Side Tracking** | **Positive:** Is it a tracking shot with the camera leading the subject from a front-side angle? | **Positive:** A tracking shot where the camera leads the subject from a front-side angle. |
| | **Negative:** Is the camera not leading the subject from a front-side angle in a tracking shot? | **Negative:** The camera is not leading the subject from a front-side angle in a tracking shot. |
| **Rear-Side Tracking** | **Positive:** Is it a tracking shot with the camera following behind the subject at a rear-side angle? | **Positive:** A tracking shot where the camera follows behind the subject at a rear-side angle. |
| | **Negative:** Is the camera not following behind the subject at a rear-side angle? | **Negative:** The camera is not following behind the subject at a rear-side angle. |
| **Lead Tracking** | **Positive:** Is it a tracking shot with the camera moving ahead of the subject as they move? | **Positive:** A tracking shot where the camera moves ahead of the subject as they move. |
| | **Negative:** Is the camera not moving ahead of the subject in a tracking shot? | **Negative:** The camera is not moving ahead of the subject in a tracking shot. |
| **Tail Tracking** | **Positive:** Is it a tracking shot with the camera following behind the subject as they move? | **Positive:** A tracking shot where the camera moves behind the subjects as they move. |
| | **Negative:** Is the camera not following behind the subject in a tracking shot? | **Negative:** The camera is not following behind the subject in a tracking shot. |
| **Tilt Tracking** | **Positive:** Does the camera tilt to track the subjects as they move? | **Positive:** A tracking shot where the camera tilts to follow the subjects. |
| | **Negative:** Is the camera not tilting to track the subjects? | **Negative:** The camera is not tilting to track the subjects. |
| **Pan Tracking** | **Positive:** Does the camera pan to track the subjects as they move? | **Positive:** A tracking shot where the camera pans to follow the subjects as they move. |
| | **Negative:** Is the camera not panning to track the subjects? | **Negative:** The camera is not panning to track the subjects. |
| **Side Tracking** | **Positive:** Is it a tracking shot with the camera moving from the side to follow the subject as they move? | **Positive:** A tracking shot where the camera moves from the side to follow the subject. |
| | **Negative:** Is the camera not moving from the side to track the subject? | **Negative:** The camera is not moving from the side to track the subject. |
| **Tracking Subject Larger** | **Positive:** Does the subject appear larger during the tracking shot? | **Positive:** The subject looks larger during the tracking shot. |
| | **Negative:** Does the subject being tracked not appear larger in size? | **Negative:** The subject being tracked does not appear larger in size. |
| **Tracking Subject Smaller** | **Positive:** Does the subject appear smaller during the tracking shot? | **Positive:** The subject looks smaller during the tracking shot. |
| | **Negative:** Does the subject being tracked not appear smaller in size? | **Negative:** The subject being tracked does not appear smaller in size. |

Table 26: **Only Motion Tasks**

| Tasks | Questions | Descriptions |
|---|---|---|
| **Only Zoom In** | **Positive:** Does the camera only zoom in with no other movement? | **Positive:** The camera only zooms in without any other movement. |
| | **Negative:** Does the camera not just zoom in? | **Negative:** The camera does not just zoom in. |
| **Only Zoom Out** | **Positive:** Does the camera only zoom out with no other movement? | **Positive:** The camera only zooms out without any other movement. |
| | **Negative:** Does the camera not just zoom out? | **Negative:** The camera does not just zoom out. |
| **Only Dolly In** | **Positive:** Does the camera only move forward (not zooming in) with respect to the ground? | **Positive:** The camera only moves forward (not zooming in) relative to the ground. |
| | **Negative:** Does the camera not just move forward with respect to the ground? | **Negative:** The camera does not just move forward relative to the ground. |
| **Only Dolly Out** | **Positive:** Does the camera only move backward (not zooming out) with respect to the ground? | **Positive:** The camera only moves backward (not zooming out) relative to the ground. |
| | **Negative:** Does the camera not just move backward with respect to the ground? | **Negative:** The camera does not just move backward relative to the ground. |
| **Only Pedestal Up** | **Positive:** Does the camera only move upward (not tilting up) with respect to the ground? | **Positive:** The camera only moves upward (not tilting up) relative to the ground. |
| | **Negative:** Does the camera not just move physically upward? | **Negative:** The camera does not just move physically upward. |
| **Only Pedestal Down** | **Positive:** Does the camera only move downward (not tilting down) with respect to the ground? | **Positive:** The camera only moves downward (not tilting down) relative to the ground. |
| | **Negative:** Does the camera not just move physically downward? | **Negative:** The camera does not just move physically downward. |
| **Only Truck Right** | **Positive:** Does the camera only move rightward without any other camera movements? | **Positive:** The camera only moves rightward without any other camera movements. |
| | **Negative:** Does the camera not just move laterally to the right? | **Negative:** The camera does not just move laterally to the right. |
| **Only Truck Left** | **Positive:** Does the camera only move leftward without any other camera movements? | **Positive:** The camera only moves leftward without any other camera movements. |
| | **Negative:** Does the camera not just move laterally to the left? | **Negative:** The camera does not just move laterally to the left. |
| **Only Pan Right** | **Positive:** Does the camera only pan rightward without any other camera movements? | **Positive:** The camera only pans rightward without any other camera movements. |
| | **Negative:** Does the camera not just pan right? | **Negative:** The camera does not just pan right. |
| **Only Pan Left** | **Positive:** Does the camera only pan leftward without any other camera movements? | **Positive:** The camera only pans leftward without any other camera movements. |
| | **Negative:** Does the camera not just pan left? | **Negative:** The camera does not just pan left. |
| **Only Tilt Up** | **Positive:** Does the camera only tilt upward without any other camera movements? | **Positive:** The camera only tilts upward without any other camera movements. |
| | **Negative:** Does the camera not just tilt up? | **Negative:** The camera does not just tilt up. |
| **Only Tilt Down** | **Positive:** Does the camera only tilt downward without any other camera movements? | **Positive:** The camera only tilts downward without any other camera movements. |
| | **Negative:** Does the camera not just tilt down? | **Negative:** The camera does not just tilt down. |
| **Only Roll CW** | **Positive:** Does the camera only roll clockwise without any other camera movements? | **Positive:** The camera only rolls clockwise without any other camera movements. |
| | **Negative:** Does the camera not just roll clockwise? | **Negative:** The camera does not just roll clockwise. |
| **Only Roll CCW** | **Positive:** Does the camera only roll counterclockwise without any other camera movements? | **Positive:** The camera only rolls counterclockwise without any other camera movements. |
| | **Negative:** Does the camera not just roll counterclockwise? | **Negative:** The camera does not just roll counterclockwise. |

