# OpenReview forum: "Towards Understanding Camera Motions in Any Video"
_NeurIPS.cc/2025/Datasets_and_Benchmarks_Track — NeurIPS 2025 Datasets and Benchmarks Track spotlight_

### Official Review · Reviewer_6DFx · 2025-06-25

**Rating:** 5
**Confidence:** 3

**Summary:**

The submission #1204 entitled "Towards Understanding Camera Motions in Any Video" presents a dataset and benchmark dedicated to understanding camera motions in videos. The key contribution of this paper lies in the construction of a large, systematically/well annotated dataset with cinematic camera motion labels (e.g., pan, tilt, truck etc.), applied to a few thousands of short clips. The annotations were curated by professional cinematographers, which distinguishes this dataset from prior efforts that were either synthetic or coarsely labeled.
Indeed, while camera motion datasets have been explored in the past, this work stands out through the granularity of the motion taxonomy, the size of the annotated corpus, and the professional-level rigor of the annotations. The dataset is also coupled with baseline experiments that explore several downstream tasks including motion classification and generative modeling guidance, which further reinforce the relevance of the proposed benchmark.

**Dataset Code Accessibility:**

Yes

**Ethical Considerations:**

No, there are no or only very minor ethics concerns

**Final Justification:**

After reading the other reviewers’ remarks and considering the authors’ rebuttal, I see no reason to change my earlier evaluation.
I therefore keep my positive rating. The dataset is interesting and beneficial for the community.

**Limitations Weaknesses:**

- Some of the figures are redundant with the text and could be better used to convey new insights, such as visual summaries of motion transitions.
- It would be useful to provide a detailed statistical breakdown of annotation frequencies per class. Some motion types are likely to be underrepresented, which could affect downstream training and evaluation.
- A visual timeline or bar showing how different motions appear throughout individual videos (e.g., a motion segmentation bar) could help the reader better grasp the temporal structure of the annotations.
- SfM is used as a baseline, but it is ill-suited for this task. It is sensitive to intrinsic parameter changes, degenerate cases (e.g., pure rotation), and dynamic environments. But as you might not have anything better to compare with, I understand its inclusion, but probably optical flow would be more telling for such a classification.

**Strengths Contributions:**

- The release is well aligned with the growing interest in video generation and 3D neural rendering, where camera motion control has become quite trendy.
- The dataset includes a wide variety of motion types, some of which are rarely addressed in existing benchmarks.
- The use of professional annotators is a big plus
- The protocol is clearly documented and can serve as a reference for future efforts.
- The paper is relatively well written, and the editorial quality is solid.

---

> ### Author Rebuttal · Authors · 2025-07-30
>
> We thank Reviewer 6DFx for the thoughtful and encouraging feedback, and for highlighting the broader relevance of this work. We also appreciate your constructive suggestions to improve the presentation. Below, we respond in detail:
>
> > **Some of the figures are redundant with the text and could be better used to convey new insights, such as visual summaries of motion transitions.**
>
> Thank you—this is a great suggestion.  Since NeurIPS does not allow URLs in the rebuttal, we will include **timeline-style** visualizations in the camera-ready version and on our Hugging Face project site to show how camera motions evolve throughout a video. We will also tighten the text to avoid repetition.
>
> > **It would be useful to provide a detailed statistical breakdown of annotation frequencies per class. Some motion types are likely to be underrepresented, which could affect downstream training and evaluation.**
>
> We agree, and we now include a detailed breakdown of binary motion primitive frequencies and model AP scores in the table below. As expected, rare classes like *Roll*—which appear mostly in cinematic productions and rarely in everyday videos—are harder to predict than common ones like *Pan*.
>
> | Motion Type   | Ratio (%) | Qwen-2.5-72B-SFT AP |
> |---------------|-----------|-------------|
> | Dolly-In      | 19.3      | 86.2        |
> | Dolly-Out     | 7.4       | 51.9        |
> | Pedestal-Up   | 4.9       | 28.2        |
> | Pedestal-Down | 6.0       | 51.9        |
> | Truck-Right   | 6.0       | 65.8        |
> | Truck-Left    | 4.6       | 57.2        |
> | Zoom-In       | 4.1       | 54.3        |
> | Zoom-Out      | 3.4       | 74.1        |
> | Pan-Right     | 7.1       | 79.4        |
> | Pan-Left      | 7.7       | 89.2        |
> | Tilt-Up       | 4.8       | 73.2        |
> | Tilt-Down     | 4.6       | 62.3        |
> | Roll-CW       | 2.7       | 66.4        |
> | Roll-CCW      | 3.2       | 62.4        |
>
> This confirms the **long-tailed** nature of camera motion. We will make this challenge more explicit in the camera-ready by including per-class frequencies.
>
> > **A visual timeline or bar showing how different motions appear throughout individual videos (e.g., a motion segmentation bar) could help the reader better grasp the temporal structure of the annotations.**
>
> Great idea—we will create a timeline-style demo video to showcase this on our Hugging Face project website.
>
> > **SfM is used as a baseline, but it is ill-suited for this task. It is sensitive to intrinsic parameter changes, degenerate cases (e.g., pure rotation), and dynamic environments. But as you might not have anything better to compare with, I understand its inclusion, but probably optical flow would be more telling for such a classification.**
>
> We agree that optical flow is a valuable signal for camera motion understanding. While directly applying 2D optical flow to our tasks may not be straightforward, state-of-the-art SfM systems like MegaSAM (CVPR 2025 Best Paper) already incorporate it internally (e.g., via SEA-RAFT), which likely contributes to their strong pose estimation performance.

---

> > ### Comment · Reviewer_6DFx · 2025-08-09
> > **Thank you for the rebutall**
> >
> > Dear authors,
> > Thank you so much for your feedback and my apology for this delayed reply (I just became dad a few days ago so it was quite complicated to take fully part of this rebutall process, thank you for your kind understanding).
> > Your responses clarify very well my interrogations and after reading other reviewer’s feedback I am convinced my first evaluation is on point. This paper definitely deserves to be accepted, congratulations. It is a great work that will be beneficial to the community.

---

> > > ### Author Response · Authors · 2025-08-09
> > >
> > > Dear Reviewer 6DFx,
> > >
> > > Thank you for your thoughtful suggestions and strong support for our work, and congratulations on becoming a father — we wish you and your family all the best during this exciting time!

---

### Official Review · Reviewer_ydYV · 2025-06-29

**Rating:** 5
**Confidence:** 3

**Summary:**

This work proposes a large-scale dataset and benchmark for understanding camera motion from videos.

It introduces a detailed and expert-informed taxonomy of camera motion primitives, along with a robust annotation framework and training program to ensure high-quality data.

The study evaluates both SfM/SLAM methods and vision-language models (VLMs) on the benchmark, highlighting the strengths and limitations of existing approaches.

**Dataset Code Accessibility:**

Yes

**Ethical Considerations:**

No, there are no or only very minor ethics concerns

**Final Justification:**

I have read the rebuttal and other reviews. The rebuttal has addressed my previous concerns and questions. I am inclined to maintain my positive score and believe the dataset is very useful for the community.

**Limitations Weaknesses:**

1. It would be helpful if the authors could compare the scale of CameraBench with existing datasets. Since camera motion is manually annotated, the scalability of CameraBench may be limited.
2. Regarding the evaluation of SfM/SLAM methods: Line 247 mentions that "the seven degrees of translation, rotation, and focal change are calculated." However, it is unclear how these continuous values are used to perform binary classification of motion primitives. Additionally, is the comparison between SfM/SLAM methods and VLMs conducted in a fair and consistent manner?

**Strengths Contributions:**

1. The curated datasets include detailed annotations of camera motion, verified by experts to ensure high quality. This is particularly valuable for downstream tasks such as camera-controlled video generation.
2. The benchmark covers a wide range of state-of-the-art Structure-from-Motion (SfM) and vision-language models (VLMs), offering important insights that can inspire future research.
3. VLMs fine-tuned on the CameraBench dataset achieve the best performance on camera motion understanding tasks.
4. The paper is well-written and provides comprehensive details on the dataset curation process, its content, statistics, and benchmarking methods.

---

> ### Author Rebuttal · Authors · 2025-07-30
>
> We thank reviewer ydYV for your positive feedback! We address your questions below:
>
> > **It would be helpful if the authors could compare the scale of CameraBench with existing datasets. Since camera motion is manually annotated, the scalability of CameraBench may be limited.**
>
> We compare CameraBench to recent video-language benchmarks in the table below:
>
> | Dataset               | Year | Videos | QA Pairs | Captions | Tags |
> |-----------------------|------|--------|----------|----------|------|
> | DREAM-1K [1]          | 2024 | 1k     | N/A      | 1k       | N/A  |
> | VDC [2]               | 2024 | 1k     | N/A      | 1k       | N/A  |
> | ActionAtlas [3]       | 2024 | 0.9k   | 0.9k     | N/A      | N/A  |
> | Vinoground [4]        | 2024 | 1k     | N/A      | 1k       | N/A  |
> | Video-MME [5]         | 2024 | 0.9k   | 2.7k     | N/A      | N/A  |
> | MovieNet [6]          | 2020 | 46k    | N/A      | 12k      | 4    |
> | AVE [7]               | 2022 | 196k   | N/A      | N/A      | 5    |
> | VidComposition [8]    | 2024 | 1k     | 1.7k     | 1.7k     | N/A  |
> | **CameraBench (total)**     | 2025 | 3.3k   | 98k      | 3.3k     | 50   |
> | **CameraBench (test-only)** | 2025 | 1k     | 10k      | 1k       | 50   |
>
> These comparisons are **not apples-to-apples**, as each dataset targets different objectives (e.g., temporal reasoning in Vinoground, human sports in ActionAtlas). CameraBench is the only dataset specifically designed for camera motion.
>
> CameraBench’s testset is intentionally sized to match recent video-language **evaluation** benchmarks (~1k videos for testing). We also release the full dataset—including training videos—to support future research in this direction.
>
> Lastly, among other datasets that include camera motion (AVE, VDC, VidComposition, Dream1K), we found major gaps in motion coverage and serious quality issues. Most define only 4–5 coarse motion types (compared to CameraBench’s 50), and our manual review of these four datasets revealed label error rates as high as 50–60%, likely due to vague annotation guidelines and no quality control. We encourage you to explore the detailed error reports linked in Supplement Section A (*Error Analysis of Prior Datasets*).
>
> > **Regarding the evaluation of SfM/SLAM methods: Line 247 mentions that "the seven degrees of translation, rotation, and focal change are calculated." However, it is unclear how these continuous values are used to perform binary classification of motion primitives. Additionally, is the comparison between SfM/SLAM methods and VLMs conducted in a fair and consistent manner?**
>
> Thank you for raising this. We clarify below:
>
> For binary classification, we follow standard practice and report Average Precision (AP). Each binary classifier outputs a *continuous* score (i.e., confidence score) indicating how likely a motion primitive is present. AP is then computed based on the ranking of these scores across all positive and negative samples.
>
> To obtain a continuous score from SfM outputs, we compute the relative translation, rotation, and focal length change between the first and last frame. See Supplement Section E (*Experimental Setup and Results*), Lines 704–708 for implementation details.
> For discriminative VLMs (e.g., CLIP-based), we compute the dot product (i.e., CLIPScore) between the video embedding and a text prompt describing the motion (e.g., “The camera is tilting up”).
>
> For generative VLMs (e.g., Qwen-2.5), we ask templated questions (e.g., “Is the camera tilting up?”) and use the model’s confidence score, i.e., P(“Yes”), as the prediction.
>
> All models are evaluated using the same set of binary motion labels and the same evaluation protocol. Additional model-specific details are provided in Supplement Section E, and we will release all evaluation code for full reproducibility.
>
> **References**:
>
> [1] Tarsier: Recipes for training and evaluating large video description models. Wang et al. 2024.
>
> [2] Auroracap: Efficient, performant video detailed captioning and a new benchmark. Chai et al. 2024.
>
> [3] ActionAtlas: A VideoQA Benchmark for Domain-specialized Action Recognition. Salehi et al. 2024.
>
> [4] Vinoground: Scrutinizing LMMs over Dense Temporal Reasoning with Short Videos. Cai et al. 2024.
>
> [5] Video-MME: The First-Ever Comprehensive Evaluation Benchmark of Multi-modal LLMs in Video Analysis. Fu et al. 2024.
>
> [6] MovieNet: A Holistic Dataset for Movie Understanding. Huang et al. 2020.
>
> [7] The Anatomy of Video Editing: A Dataset and Benchmark Suite for AI-Assisted Video Editing. Argaw et al. 2022.
>
> [8] VidComposition: Can MLLMs Analyze Compositions in Compiled Videos? Tang et al. 2024.

---

### Official Review · Reviewer_owMA · 2025-06-30

**Rating:** 5
**Confidence:** 4

**Summary:**

This paper introduces CameraBench, a large-scale dataset and benchmark for understanding camera motion in videos. The authors introduce taxonomy of camera motion primitives and design a robust annotation framework. They demonstrate that fine-tuning a generative VLM on CameraBench significantly improves its performance on tasks like motion-augmented captioning and video question answering, showcasing the benchmark's value for future model development.

**Additional Feedback:**

N/A

**Dataset Code Accessibility:**

Yes

**Dataset Code Comments:**

The paper provides a project website URL that directs to the dataset, code, and documentation.

**Ethical Comments:**

The videos in the proposed dataset are sourced from YouTube, and the authors state they adhere to educational licenses. They acknowledge the potential for generative models to be misused and discuss safeguards in the supplementary material.

**Ethical Considerations:**

No, there are no or only very minor ethics concerns

**Final Justification:**

After reading the rebuttal and other reviews, my assessment remains positive. The authors have effectively addressed my concerns, and I believe the proposed dataset will be a valuable asset to the community. Thus, I am maintaining my score.

**Limitations Weaknesses:**

1. Dataset scale for training: The training set consists of ~1,400 videos. While the authors emphasize its high quality, this is a relatively small number for training large foundation models. The model's ability to generalize to camera motions, genres, or compositions that are rare or absent in this training set remains an open question.

2. Annotation of long duration and complex sequences: The dataset consists of manually segmented single shots with an average duration of 5.7 seconds. While this ensures clean annotation of individual primitives, real-world video often involves much longer, unsegmented clips with multiple complex shot transitions.

3. Temporal ordering of camera motion: The proposed “label-then-caption” approach with the annotation framework seems not to available to capture sequential motions (e.g., “The camera first pans left, then right”). The lack of structured temporal information could limit the development of models that aim to understand or generate precise, ordered sequences of camera movements.

**Strengths Contributions:**

1. Rigorous taxonomy of camera motion: The core contribution is the development of a detailed taxonomy of camera motion, created in collaboration with professional cinematographers. This taxonomy is far more comprehensive than those in prior work (which often have only 4-5 categories ), using precise terminology to distinguish between easily confused concepts like translation (dolly/truck), rotation (pan/tilt), and intrinsic changes (zoom). It also introduces multiple frames of reference (camera, ground, object) to resolve ambiguity.

2. Carefully designed annotation process: The taxonomy was developed with expert collaboration and iterative refinements. Also, the “label-then-caption” approach is improves the robustness of annotations for real-world videos.

3. Comprehensive benchmarking and analysis: The authors evaluate a diverse suite of 20 modern models, including classic and learning-based SfM/SLAM methods and various VLMs. They also fine-tune a VLM on the CameraBench and validate the utility of the proposed dataset for model development.

---

> ### Author Rebuttal · Authors · 2025-07-30
>
> We thank reviewer owMA for your positive comments! We address your questions below:
>
> > **The training set consists of 1,400 videos. While the authors emphasize its high quality, this is a relatively small number for training large foundation models.**
>
> Thank you for recognizing the quality of our dataset and encouraging us to expand it. Our main goal is to **evaluate** foundation models on camera motion understanding, rather than train them from scratch. CameraBench includes over 3,000 videos in total—comparable in scale to other evaluation benchmarks like Dream1K [1], VDC [2], ActionAtlas [3], and Vinoground [4], which each contain around 1,000-2,000 videos.
>
> What sets CameraBench apart is the **richness of annotations**: each video is labeled with 50+ motion primitives, paired with carefully written descriptions, and passes through multiple rounds of expert review. Given the high cost per video, we scaled as much as possible within our academic budget. We see the fine-tuning on a modest training set (~1,400 videos) as a promising result, showing that even a limited amount of high-quality supervision can lead to meaningful improvements.
>
> > **The model's ability to generalize to camera motions, genres, or compositions that are rare or absent in this training set remains an open question.**
>
> Thank you for raising this important point. Below are label frequency ratios and Qwen-2.5-72B AP scores across 14 motion primitives in the camera-centric frame. As expected, common motions (e.g., Dolly-In) achieve higher accuracy than rare ones (e.g., Roll-Clockwise), suggesting that expanding labels could improve performance on long-tailed classes:
>
> | Motion Type   | Ratio (%) | Qwen-2.5 AP |
> |---------------|-----------|-------------|
> | Dolly-In      | 19.3      | 86.2        |
> | Dolly-Out     | 7.4       | 51.9        |
> | Pedestal-Up   | 4.9       | 28.2        |
> | Pedestal-Down | 6.0       | 51.9        |
> | Truck-Right   | 6.0       | 65.8        |
> | Truck-Left    | 4.6       | 57.2        |
> | Zoom-In       | 4.1       | 54.3        |
> | Zoom-Out      | 3.4       | 74.1        |
> | Pan-Right     | 7.1       | 79.4        |
> | Pan-Left      | 7.7       | 89.2        |
> | Tilt-Up       | 4.8       | 73.2        |
> | Tilt-Down     | 4.6       | 62.3        |
> | Roll-CW       | 2.7       | 66.4        |
> | Roll-CCW      | 3.2       | 62.4        |
>
> CameraBench also improves on prior work by covering a broader range of genres. While other datasets like MovieNet and AVE focus solely on film, CameraBench includes documentary (25%), gaming (11%), anime (5%), vlogs (9%), and more—better reflecting the diversity of internet video.
>
> > **The proposed “label-then-caption” approach with the annotation framework seems not to available to capture sequential motions (e.g., “The camera first pans left, then right”)**
>
> This seems to be a misunderstanding—our approach *does* support sequential motions like this. In cases like “the camera pans left then right”, the annotator selects “*complex-motion*” to indicate conflicting directions and does **not** check any panning labels such as “pan-left,” “pan-right,” or “no-pan”. Instead, they write a text caption to capture the temporal order of the two motions.
>
> In fact, most CameraBench captions describe more than two sequential motions, such as:
>
> - "The camera smoothly trucks slightly to the left, then quickly tilts downward before moving backward to follow the skateboarder, maintaining minimal shaking throughout."
> - "The drone smoothly glides forward over the cityscape, tilting down to capture the scene below. Midway, it gradually pans left to adjust its orientation, maintaining a steady and fluid motion throughout."
> - "The unsteady camera quickly pans from left to right, then moves forward to approach the screen, with a slight shake throughout the rapid movement."
>
> Please visit our project site and Hugging Face page for more video examples.
>
> While we use natural language to flexibly describe complex, sequential camera motion, future work may explore temporal segmentation of motion primitives—though this remains challenging, as each short clip can contain 1–10+ motions, and boundaries can be ambiguous for some motion types like tracking shots.
>
> > **Real-world video often involves much longer, unsegmented clips with multiple complex shot transitions.**
>
> CameraBench can potentially support studying longer clips: on Hugging Face, we provide the start and end timestamps of each clip from the original video, so users can combine per-clip annotations if they want to analyze camera motion over longer sequences. We leave this challenge to future work that builds on CameraBench.
>
> **References**:
>
> [1] Tarsier: Recipes for training and evaluating large video description models. Wang et al. 2024.
>
> [2] Auroracap: Efficient, performant video detailed captioning and a new benchmark. Chai et al. 2024.
>
> [3] ActionAtlas: A VideoQA Benchmark for Domain-specialized Action Recognition. Salehi et al. 2024.
>
> [4] Vinoground: Scrutinizing LMMs over Dense Temporal Reasoning with Short Videos. Cai et al. 2024.

---

> > ### Author Response · Authors · 2025-08-05
> >
> > Dear Reviewer owMA,
> >
> > As this is the last day of discussion, we hope our responses have addressed your concerns. If anything remains unclear, please let us know (we’ll do our best to reply today); otherwise, we’d appreciate it if you could consider updating your score.

---

### Official Review · Reviewer_Hh66 · 2025-07-01

**Ethics Flags:** Data privacy, copyright, and consent,…
**Rating:** 5
**Confidence:** 3

**Summary:**

This paper presents CameraBench, a large-scale dataset and benchmark for evaluating camera motion understanding in unconstrained internet videos. Collaborating with professional cinematographers, the authors develop a fine-grained taxonomy of 50+ motion primitives and a multi-stage annotation pipeline involving expert training and quality control. The proposed CameraBench spans 3,381 videos with over 150K expert-curated binary labels and captions, supporting classification, retrieval, VQA, and captioning tasks. The authors benchmark 20+ SfM/SLAM and VLM models and show that fine-tuning Qwen2.5-VL significantly enhances motion understanding performance.

**Additional Feedback:**

- 1. I would appreciate clarification on whether motion labels in the dataset are mutually exclusive or can co-occur. For example, can a single shot be labeled with both dolly-in and tilt-up, or are annotators restricted to selecting only one motion type per category?

- 2. It would be helpful to report annotator agreement metrics (e.g., Fleiss' Kappa or Krippendorff's Alpha) for key motion primitives. Given the fine-grained nature of the taxonomy and the subjective aspects involved, this would provide more confidence in label reliability.

- 3. Section 5 could benefit from a deeper analysis of model failure modes. Specifically, it would be interesting to know which types of motion primitives tend to challenge SfM/SLAM systems versus VLMs. Are there systematic weaknesses (e.g., with roll, zoom, or tracking shots) that correlate with model architecture or input modality?

**Dataset Code Accessibility:**

No

**Ethical Comments:**

- Use of internet videos with identifiable subjects raises privacy concerns. Are subjects (e.g., in selfies or family videos) aware their footage is being annotated and benchmarked?

- The dataset does not provide any terms of use or license. It’s unclear whether downstream models trained on CameraBench can be used commercially.

- Potential for misuse in video surveillance, behavior tracking, or profiling is not addressed at all.

**Ethical Considerations:**

Yes, there are ethics concerns that require attention by the authors

**Final Justification:**

After carefully reading the authors’ rebuttal and supplementary analyses, I appreciate the substantial effort they put into addressing the concerns raised in my initial review. The authors clarified several points, added useful results, and demonstrated a strong commitment to transparency and annotation quality.

As for the weakness part:
- While I now better understand the challenges of evaluating SfM/SLAM systems on unconstrained internet video, I still believe the current evaluation setup inherently favors VLMs, particularly those fine-tuned on the benchmark. CameraBench would benefit from clearer baseline definitions and alternative evaluation protocols tailored to geometric models.

- The lack of broader transfer results in the main paper is still a weakness. Although addressed in the rebuttal and supplement, this information would be more impactful if foregrounded in the primary submission.

- Ethical concerns related to video usage, consent, and dataset licensing were adequately addressed in the rebuttal, and I appreciate the authors’ plans to make takedown mechanisms and intended use clearer in the final version.

**Limitations Weaknesses:**

While I appreciate the ambition and scope of this work, I have several serious concerns regarding the evaluation design, fairness， as detailed as below:

1. One of my main concerns is that the evaluation heavily favors vision-language models (VLMs), especially those compatible with fine-tuning, while providing no meaningful opportunity for traditional SfM/SLAM methods to compete.
The authors devote extensive effort to fine-tuning Qwen2.5-VL (with ∼1400 videos), then compare it against out-of-the-box SfM pipelines like COLMAP and MegaSAM on tasks that are clearly not designed for them, such as VQA and captioning. As far as I can tell, SfM models are only evaluated based on a binary motion primitive classifier derived from first/last-frame pose deltas, which is an extremely limiting and lossy summary of their capabilities.
2. As for the experiments, the fine-tuned Qwen2.5-VL models are only evaluated on the held-out portion of CameraBench — a dataset that shares the same annotation style, taxonomy, and video source.
There is no effort to test generalization to other datasets (e.g., MovieNet, AVE, VDC, or even video captioning benchmarks like ActivityNet or Ego4D). This makes the gains from supervised fine-tuning look more like overfitting to the benchmark than genuine progress toward motion understanding.
I fail to see why the authors didn’t evaluate transfer or zero-shot behavior on related datasets. If this work claims to push “understanding camera motion in any video,” some generalization analysis is essential.

**Strengths Contributions:**

I think this is one of the most ambitious and comprehensive attempts to benchmark camera motion understanding to date. Several aspects stand out clearly to me:

- 1.The proposed taxonomy of over 50 camera motion primitives is exceptionally detailed. As far as I can tell, this is the first dataset that not only clearly separates camera-centric, ground-centric, and object-centric frames of reference, but also provides direction-specific annotations (e.g., pan-left vs. pan-right), which many existing datasets ignore or conflate. Figure 3 summarizes this taxonomy effectively, and its design is grounded in professional cinematographic practice.

- 2. I appreciate the rigor behind the annotation process. Rather than relying on crowdworkers with minimal guidance, the authors  implement a label-then-caption strategy that addresses ambiguity by allowing annotators to skip uncertain labels and provide contextual descriptions instead. Furthermore, their training pipeline includes lecture-based instruction, five rounds of performance feedback, and expert consensus audits, as described in Section 4 and Figure 4.

---

> ### Author Rebuttal · Authors · 2025-07-30
>
> We sincerely thank the reviewer Hh66 for recognizing our fine-grained taxonomy and rigorous annotation pipeline. We respond to your thoughtful comments below:
>
> > **Current evaluation favors VLMs, with no meaningful opportunity for SfM/SLAM methods to compete.**
>
> We agree that CameraBench currently favors VLMs—this reflects a broader challenge in the field: **modern SfM systems cannot be end-to-end fine-tuned on in-the-wild internet videos due to the lack of ground-truth camera trajectories**, which require specialized camera equipment. Rather than a flaw, we see this as an important gap that CameraBench helps expose.
>
> To explore this direction, we trained simple 3-layer MLP classifiers on top of MegaSAM-generated pose trajectories (using 7D pose deltas across frames) with logistic regression to predict binary motion primitives using CameraBench's training split. As shown below, this lightweight supervision, which incorporates temporal information beyond just the first and last frame, improves MegaSAM’s performance:
>
> | Motion Type | MegaSAM (zero-shot) | MegaSAM (SFT) | Qwen-2.5 (zero-shot) | Qwen-2.5 (SFT) |
> | ------------- | ------------------- | ------------- | -------------------- | -------------- |
> | Dolly-In | 73.8 | 77.0 | 63.0 | 83.2 |
> | Dolly-Out | 43.9 | 48.9 | 14.1 | 48.6 |
> | Pedestal-Up | 24.2 | 36.6 | 20.1 | 27.2 |
> | Pedestal-Down | 29.1 | 42.8 | 22.3 | 48.8 |
> | Truck-Right | 45.3 | 48.1 | 28.5 | 62.6 |
> | Truck-Left | 44.2 | 47.5 | 27.7 | 54.3 |
> | Zoom-In | 11.1 | 16.1 | 23.2 | 51.3 |
> | Zoom-Out | 10.2 | 14.2 | 27.2 | 70.7 |
> | Pan-Right | 79.5 | 84.1 | 36.5 | 77.6 |
> | Pan-Left | 82.2 | 87.4 | 44.6 | 86.9 |
> | Tilt-Up | 73.8 | 79.9 | 38.4 | 70.4 |
> | Tilt-Down | 65.3 | 68.6 | 25.7 | 58.0 |
> | Roll-CW | 71.5 | 74.3 | 26.0 | 38.5 |
> | Roll-CCW | 75.8 | 77.6 | 25.5 | 46.3 |
> | Static | 22.0 | 59.6 | 20.2 | 65.2 |
> | **Average** | **50.1** | **57.5** | **29.5** | **59.3** |
>
> These results show that CameraBench can provide useful training signals for SfMs, even without ground-truth trajectories. We hope this encourages further work on adapting SfM to large-scale internet video using scalable supervision.
>
> > **SfM is not designed for VQA or captioning.**
>
> Current SfM methods are not designed for language tasks like VQA or captioning, so we do not evaluate them on those tasks. However, one long-term goal of CameraBench is to encourage future work that bridges SfM and VLMs for video-language understanding.
>
> > **Binary motion classification is a lossy summary of SfM capabilities.**
>
> While binary motion prediction is indeed a simplified view of SfM outputs, we see it as a critical **sanity check**: if an SfM method truly estimates accurate trajectories, it should handle basic motion classification (e.g., dolly-in vs. not). The fact that many SfMs struggle here underscores the need for a benchmark like CameraBench to diagnose such failures and guide progress.
>
> > **Can CameraBench-finetuned Qwen-2.5 generalize better to other motion understanding datasets (e.g., MovieNet, AVE, VDC, or even video captioning benchmarks like ActivityNet or Ego4D)?**
>
> Yes! As shown below, CameraBench-finetuned Qwen-2.5-7B improves average per-class accuracy by up to 5% on AVE’s val-test set (ECCV 2022):
>
> | Class | Qwen-2.5 (SFT) | Qwen-2.5 (zero-shot) | Prev. SOTA in paper (Logit Adjustment) |
> | ----------------- | -------------- | -------------------- | ---------------------------- |
> | handheld | 0.69 | 0.56 | 0.27 |
> | locked | 0.71 | 0.61 | 0.82 |
> | pan/truck | 0.52 | 0.51 | 0.27 |
> | tilt/pedestal | 0.53 | 0.52 | 0.44 |
> | zoom/dolly | 0.53 | 0.51 | 0.40 |
> | **Mean Accuracy** | **0.59** | **0.54** | **0.44** |
>
> However, AVE is limited to only five coarse motion types: pan/truck, tilt/pedestal, locked, zoom/dolly, and handheld. In contrast to CameraBench’s 50+ fine-grained labels, AVE does not capture motion direction and often conflates intrinsic vs. extrinsic changes (e.g., dolly vs. zoom) as well as rotation vs. translation (e.g., tilt vs. pedestal). See Line 538 for further discussion on AVE’s limitations.
>
> We also observe a significant improvement on VDC’s Camera Split (using their official evaluation protocol) after CameraBench fine-tuning:
>
> | Model | Accuracy (%) | LLM Eval Score |
> | ------------------------- | ------------ | -------------- |
> | Qwen-2.5 (zero-shot) | 38.8 | 1.75 |
> | Qwen-2.5 (SFT) | 57.6 | 2.64 |
> | Prev. SOTA in paper (AuroraCap-7B) | 42.5 | 2.27 |
>
> That said, VDC suffers from serious quality issues when it comes to camera motion. In Supplement Section A (*Error Analysis of Prior Datasets*), we provide website links for readers to view these errors directly.
>
> Finally, ActivityNet and Ego4D focus on human motion (e.g., hand-object interactions and daily activities), which is outside the scope of CameraBench’s focus on camera motion.
>
> > **Why didn’t the authors evaluate transfer or zero-shot behavior on related datasets?**
>
> We did not include these generalization results in the main paper because AVE and VDC are annotated by crowdworkers and contain **substantial label errors**. We discuss this in Supplement Section A (*Error Analysis of Prior Datasets*) and provide website links for readers to see the errors and our expert feedback directly.
>
> > **Use of internet videos raises ethical concerns (privacy, licensing, misuse).**
>
> Thank you for raising this important point. We follow standard academic practice (e.g., AVE, ActivityNet) by annotating publicly available YouTube videos for non-commercial research. These videos remain subject to YouTube’s terms of service, which allow public viewing and linking for research use.
>
> All CameraBench annotations (labels and captions) are released under a CC-BY-4.0 license. The videos themselves remain governed by YouTube’s terms and are intended strictly for research, not commercial deployment. Our Hugging Face page states that we honor takedown requests—any video will be removed within 3–7 days upon request from the video uploader. We have also contacted the creators of videos highlighted on our project site to request permission for continued research use. If any takedown requests arise, we will promptly replace those videos.
>
> CameraBench does not focus on human motion or identity, and is not suitable for tasks such as person tracking, behavior analysis, or profiling. We will clarify these potential misuse concerns in the camera-ready.
>
> > **Whether motion labels in the dataset are mutually exclusive or can co-occur. For example, can a single shot be labeled with both dolly-in and tilt-up, or are annotators restricted to selecting only one motion type per category?**
>
> Most CameraBench labels **can** co-occur. For example, Figure 5 (bottom row) shows a video annotated with multiple labels including “dolly-in,” “pedestal-down,” “pan-left,” “tilt-up,” and “arc-ccw”. You can also explore other multi-label examples on our project website.
>
> However, certain labels are mutually exclusive. For instance, no video is labeled with both “dolly-in” and “dolly-out” (which would be contradictory). If a video shows a dolly-in followed by a dolly-out, we ask annotators **not** to select any dolly labels, but instead describe both events in the language description. This annotation framework is detailed in Supplement Sections C and D, if you're interested.
>
> > **It would be helpful to report annotator agreement metrics (e.g., Fleiss' Kappa or Krippendorff's Alpha) for key motion primitives. Given the fine-grained nature of the taxonomy and the subjective aspects involved, this would provide more confidence in label reliability.**
>
> You are right—this task is challenging due to the fine-grained nature of our taxonomy. Instead of using the majority vote of crowdworkers and reporting agreement metrics, we go one step further and enforce **consensus for every annotation**. As noted in Line 235, “*we hold feedback sessions and revise annotations to reach consensus*” when there is disagreement. Also, every video in CameraBench is “*reviewed by at least one author during the quality control phase*” (Line 69).
>
> To ensure high label accuracy, we conduct rigorous annotator screening and training (Line 218): out of 100 applicants, only the top performers who complete all five training rounds with ~95%+ accuracy are hired.
>
> We are confident in our labels and host the dataset on Hugging Face to allow public inspection and feedback.
>
> > **Section 5 could benefit from a deeper analysis of model failure modes. Specifically, it would be interesting to know which types of motion primitives tend to challenge SfM/SLAM systems versus VLMs. Are there systematic weaknesses (e.g., with roll, zoom, or tracking shots) that correlate with model architecture or input modality?**
>
> This is a great suggestion. Some motion types, like *roll*, are especially hard for VLMs. *Roll* is a rare cinematic technique that doesn’t appear often in internet videos, so models like Qwen-2.5-SFT only achieve 43–50% accuracy. In contrast, SfM models like MegaSAM do much better (72–76%) on this class.
>
> SfMs have their own limitations. For example, MegaSAM struggles with *zoom* because it doesn’t estimate focal length, which is needed to tell apart *zoom* from *dolly*. We expect future versions of MegaSAM to improve this.
>
> We’ll add this discussion in the camera-ready to help clarify where each model type succeeds or fails.

---

### Decision · Program_Chairs · 2025-09-18

**Decision:**

Accept (spotlight)

**Comment:**

This submission received positive scores from all reviewers. Although some concerns were raised during the initial review, the authors addressed them effectively in the rebuttal. Therefore, the Area Chair recommends acceptance.